# Dale's Law Meets Geometric Brownian Motion: Multiplicative Updates for Sampling

## Abstract

Gradient descent has proven to be a powerful and effective technique for optimization in numerous machine learning applications. Recent advances in computational neuroscience have shown that learning in standard gradient descent optimization formulation is not consistent with learning in biological systems. This has opened up interesting avenues for building biologically inspired learning techniques. One such approach is inspired by Dale's law, which states that inhibitory and excitatory synapses do not swap roles during the course of learning. The resulting *exponential gradient descent* optimization scheme leads to log-normally distributed synaptic weights. Interestingly, the density that satisfies the Fokker-Planck equation corresponding to the stochastic differential equation (SDE) with geometric Brownian motion (GBM) is the log-normal density. Leveraging this connection, we start with the SDE governing geometric Brownian motion, and show that discretizing the corresponding reverse-time SDE yields a multiplicative update rule, which surprisingly, coincides with the sampling equivalent of the exponential gradient descent update founded on Dale's law. Proceeding further, we propose a new formalism for multiplicative denoising score-matching, which subsumes the loss function proposed by Hyvärinen for non-negative data. Indeed, log-normally distributed data is positive and the proposed score-matching formalism turns out to be a natural fit. This allows for training of score-based models for image data and results in a novel multiplicative update scheme for sample generation starting from a log-normal density. Experimental results on MNIST, Fashion MNIST, and Kuzushiji datasets demonstrate generative capability of the new scheme. To the best of our knowledge, this is the first instance of a biologically inspired generative model employing multiplicative updates, founded on geometric Brownian motion.

## 1 Introduction

An interesting problem in computational neuroscience is training artificial neural networks (ANNs) in a fashion that is consistent with learning and optimization seen in biological systems. Several studies [50, 36, 8, 38, 41] have confirmed that synaptic weight distributions in biological systems are log-normally distributed and that the neurons obey Dale's law [15], which states that excitatory (inhibitory) neurons stay excitatory (inhibitory) throughout the course of learning without synaptic flips. Artificial neural networks trained with gradient descent seldom obey Dale's law. Recently, Cornford et al. [12] proposed the use of exponentiated gradient descent (EGD) to train neural networks and have observed that the training is consistent with Dale's law and leads to log-normally distributed synaptic weights, in alignment with experimental findings. Exponentiated gradient descent is derived using mirror descent for a particular variant of Bregman divergence.

In this paper, we establish a concrete link between exponentiated gradient descent optimization to sampling from stochastic differential equations (SDEs) inspired by geometric Brownian motion (GBM). Whereas most diffusion modeling and sampling schemes rely on standard Brownian motion, to the best of our knowledge, this is the first instance where GBM is used. We show that the proposed framework captures the multiplicative nature of updates seen in EGD. The ability of geometric Brownian motion to model processes with proportional changes makes it an ideal candidate for developing biologically inspired generative models. For the purpose of generation, we need the underlying score function used in the reverse-time SDE, for which we develop a novel multiplicative score-matching loss. While a large body of contemporary generative modeling literature is based on SDEs with additive Gaussian noise, our novel formalism relies on an SDE that governs the forward noising process dynamics with multiplicative log-normal noise. We develop the corresponding reverse-time SDE and show that it results in a multiplicative update rule that is structurally equivalent to the exponential gradient-descent scheme Cornford et al. [12]. The multiplicative update rule obtained as a consequence of the discretization of the SDE can be used to sample from the desired distribution whose score function is learnt using a neural network. We support the theoretical developments with experimental results on MNIST [31], Fashion MNIST [60] and Kuzushiji image datasets [10].

## 1.1 Related Works

Recent developments in generative modelling employing generative adversarial networks [18], diffusion models [21], score-based models [51, 52, 54], flow-based models [40] have produced stunning examples across a variety of modalities spanning images, video, audio, etc.. In the context of diffusion models, a seminal contribution has been the early work by Sohl-Dickstein et al. [47]. Inspired by non-equilibrium thermodynamics, they introduced the diffusion probabilistic model as a tractable and flexible model for sampling and inference. They demonstrated generative capability on toy datasets in two dimensions and image datasets like binarized MNIST and CIFAR-10. Ho et al. [21] demonstrated that denoising diffusion probabilistic models (DDPMs) could be used for high quality image synthesis. They vastly improved the results from Sohl-Dickstein et al. [47] and showed a performance comparable to state-of-the-art generative models [26, 27] of that time. Progress in score-matching by Song et al. [53], Song and Ermon [51, 52] demonstrated the potential of score-based generative models to be competitive with diffusion models. In the seminal work of Song et al. [55], it was shown that an SDE framework unifies both approaches. These SDEs were based on standard Brownian motion. Several alternative formulations that obviate the need for Brownian motion were also proposed. In particular, Bansal et al. [2] propose generative models that are based on more generic degradation operations and their corresponding restoration operations. They consider blurring and masking among others as degradation operators and show that such generalized degradations could also be used to formulate generative models. Rissanen et al. [43] proposed that generation could be viewed as the time-reversal of a heat equation. Additionally, they showed that their approach allows for certain image properties like shape and colour to be disentangled and they also discuss spectral properties that reveal inductive biases in generative models. Santos et al. [45] developed a discrete state-space diffusion model that relies on a pure-death random process and demonstrate competitive generative ability on binarized MNIST, CIFAR-10, and CelebA-64 datasets.

A recent preprint on image denoising by Vuong and Nguyen [57] is perhaps the closest to the multiplicative noise model considered in this paper. They consider a forward process where images are corrupted by multiplicative log-normal or gamma distributed noise. However, instead of proceeding with the multiplicative noise model, they convert it to an additive one by applying a logarithmic transformation. While the log-transformation simplifies the calculations, it reduces the problem to the additive noise setting, losing out completely on the richness of the original multiplicative noise framework. Vuong and Nguyen [57] remark explicitly that the reverse-time SDE in the multiplicative noise setting comes with a lot of complications, which are overcome by converting it to an additive noise model. They also restrict the scope of their work to denoising and do not propose a generative framework.

## 1.2 Organization of the paper

Section 2 gives an account of Dale's law and progress in computational neuroscience in deploying exponentiated gradient descent to enforce Dale's law – all of these form the inspiration for this work. In Section 3, we present the essential mathematics behind SDEs and generative modeling

required for understanding the contributions of this paper. Section 4 introduces Geometric Brownian Motion (GBM) and its corresponding reverse-time SDE based sampler for image generation. This necessitates a new score-matching framework for multiplicative noise which we define in Section 5. Finally, Section 6 presents experiments on MNIST, Fashion-MNIST, and Kuzushiji MNIST datasets, demonstrating the effectiveness and potential of the proposed model.

## 2 Dale's Law and Exponentiated Gradients

In computational neuroscience, Dale's law [15] has been empirically observed to hold in many biological systems barring certain exceptions. Dale's law states that presynaptic neurons can only *exclusively* affect their corresponding postsynaptic counterparts in an excitatory or inhibitory manner. The implication of the law is that the synapses continue to be inhibitory or excitatory during the course of learning without flipping. On the contrary, artificial neural networks have synaptic weights that can flip from excitatory to inhibitory or vice versa during training. Previous attempts [4, 58, 34] to incorporate biologically inspired learning rules to train neural networks have had limited success on standard benchmark tasks. Recently, Cornford et al. [11] demonstrated that ANNs that obey Dale's law, which they name Dale's ANNs (DANNs), can be constructed without loss in performance compared to weight updates done using standard gradient descent. They show that the ColumnEI models proposed by Song et al. [48] are suboptimal and can potentially impair the ability to learn by limiting the solution space of weights. DANNs outperform ColumnEI models on tasks across MNIST [31], Fashion-MNIST [60] and Kuzushiji MNIST datasets [10]. Cornford et al. [11] posit that the emergence and prevalence of Dale's law in biological systems is a possible evolutionary local minima and that the presence of inhibitory units in learning could help avoid catastrophic forgetting [3].

Li et al. [32] demonstrated that methods such as ColumnEI proposed by Song et al. [48] to incorporate Dale's law into the training of recurrent neural networks (RNNs) lead to suboptimal performance on sequence learning tasks, which is primarily attributed to poor spectral properties of the weight matrices, in particular, the multimodal, dispersed nature of the singular value spectrum of the weight matrix. Li et al. [32] extended the architecture developed by Cornford et al. [11] to handle sequences using RNNs and showed that these networks are on par with RNNs that are trained without incorporating Dale's law. The spectral properties of DANN RNNs are also better than the ColumnEI networks and the singular value spectrum is unimodal and clustered leading to superior performance on tasks such as the adding problem [22], sequential MNIST task [30] and language modelling using the Penn Tree Bank [37].

Cornford et al. [12] demonstrated that gradient descent is a suboptimal phenomenological fit to learning experiments in biologically relevant settings. While stochastic gradient descent for training ANNs is an exceptionally successful and robust model in general, it violates Dale's law [15] by allowing for synaptic flips. This leads to the distribution of weights not being log-normal, which contradicts experimentally observed data. Cornford et al. [12] showed that exponentiated gradient descent (EGD) introduced by Kivinen and Warmuth [29] respects Dale's law and consequently produces log-normally distributed weights. In experiments performed on the Mod-Cog framework [28] using RNNs, EGD outperforms gradient descent and is superior to GD for synaptic pruning. The learning task is formulated utilizing the mirror descent framework [39, 7] as changes to synaptic weights in a neural network such that a combination of task error and "synaptic change penalty" must be minimized. This leads to the update rule:

$$\boldsymbol{X}_{k+1} = \arg\min_{\boldsymbol{X}} \left[ \bar{\ell}(\boldsymbol{X}) + \frac{1}{\eta} D_\phi(\boldsymbol{X}, \boldsymbol{X}_k) \right], \tag{1}$$

where $\bar{\ell}(\boldsymbol{X}) = \ell(\boldsymbol{X}_k) + \nabla\ell(\boldsymbol{X})^\top |_{\boldsymbol{X}=\boldsymbol{X}_k} (\boldsymbol{X} - \boldsymbol{X}_k)$ is the linearization of the task error $\ell(\boldsymbol{X})$ about the point $\boldsymbol{X}_k$ and $D_\phi(\boldsymbol{X}, \boldsymbol{X}_k)$ is the synaptic change penalty. The penalty $D_\phi : \mathbb{R}^d \times \mathbb{R}^d \to \mathbb{R}$ is chosen as the Bregman divergence corresponding to a strictly convex function $\phi : \mathbb{R}^d \to \mathbb{R}$. Depending on the choice of $\phi$, we get different update rules. For instance, when $\phi(\boldsymbol{X}) = \|\boldsymbol{X}\|_2^2$, the corresponding synaptic change penalty is $D_\phi(\boldsymbol{X}, \boldsymbol{X}_k) = \|\boldsymbol{X} - \boldsymbol{X}_k\|_2^2$, and Eq. (1) results in the familiar gradient-descent update $\boldsymbol{X}_{k+1} = \boldsymbol{X}_k - \eta\nabla\ell(\boldsymbol{X}) |_{\boldsymbol{X}=\boldsymbol{X}_k}$. This update rule for the weights does not guarantee that the entries of $\boldsymbol{X}_{k+1}$ and $\boldsymbol{X}_k$ have the same sign, which allows for synaptic flips during training, as also confirmed by Cornford et al. [12].

Cornford et al. [12] chose $\phi(\boldsymbol{X}) = \sum_{i=1}^{d} |X^{(i)}| \log |X^{(i)}|$, where $X^{(i)}$ denotes the $i^{th}$ entry of $\boldsymbol{X}$, which results in $D_\phi$ being the unnormalised relative entropy,

$$D_\phi(\boldsymbol{X}, \boldsymbol{X}_k) = \sum_{i=1}^{d} X^{(i)} \log \frac{X^{(i)}}{X_k^{(i)}} - X^{(i)} + X_k^{(i)}.$$

For this choice of $D_\phi$, the update rule in Eq. (1) takes the form

$$\boldsymbol{X}_{k+1} = \boldsymbol{X}_k \circ \exp\left(-\eta \nabla_{\boldsymbol{X}} \ell(\boldsymbol{X}) \mid_{\boldsymbol{X}=\boldsymbol{X}_k} \circ \operatorname{sign}(\boldsymbol{X}_k)\right), \tag{2}$$

where $\circ$ denotes element-wise multiplication. The update in Eq. 2 is different from standard gradient-descent update in many ways: the update is multiplicative as opposed to additive, involves exponentiation, and preserves the sign of the entries of $\boldsymbol{X}_k$ as iterations proceed. Effectively, the entries in $\boldsymbol{X}_k$ for any $k$ have the same sign as those in $\boldsymbol{X}_0$. The update rule in Eq. (2) is referred to as *exponentiated gradient descent (EGD)* [29].

By design, EGD doesn't allow synaptic flips and automatically respects Dale's law during the course of training. The update rule also leads to the weights being distributed log-normally as demonstrated by Pogodin et al. [41]. Exponentiated gradient-descent has been shown to perform on par with gradient descent for models trained on Mod-Cog tasks, although the final weight distributions are different. The networks for both updates are initialized with log-normal weights to adhere to experimental data that shows that the synaptic strengths of neurons in the brain are log-normally distributed [14, 35]. The network trained with gradient descent had a final weight distribution that was different from log-normal whereas the network trained with exponentiated gradient was log-normally distributed. Additionally, Cornford et al. [12] have shown that learning with EGD is more robust to synaptic weight pruning and EGD outperforms gradient descent when relevant inputs are sparse and in particular, for continuous control tasks. Pogodin et al. [41] showed that the distribution of converged weights depends on the geometry induced by the choice of the update algorithm. Gradient-descent updates implicitly assume Euclidean geometry, which is inconsistent with the log-normal weight distribution that is experimentally observed and is ill-suited to data arising in neuroscience.

A quick glance at Eq. (2) prompts the question: **Does there exist a sampling equivalent for the exponentiated gradient-descent update rule?** This is inspired by the link between gradient-descent and Langevin dynamics as enunciated by Wibisono [59]. In pursuit of an answer to this question, we realised the connection between the log-normally distributed weights observed at the end of exponentiated gradient descent and the sampling equation lies in geometric Brownian motion. The equilibrium distribution of GBM is the log-normal density and its time-reversal would give us the sampling formula we seek (discussed in Sec. 4).

## 3 Stochastic Differential Equations and Generative Modelling

Recent generative models such as diffusion models [21, 49] and score-based models rely heavily on the SDE framework. These models have been immensely successful in generating realistic samples across different data modalities such as images [55] and audio [42]. The key idea is to construct a stochastic process such that one starts with samples from the true, unknown density and progressively transforms them to samples from a noisy, easy-to-sample-from density such as the isotropic Gaussian. The task of generation requires inverting the forward process which goes beyond mere time reversal due to the stochastic nature of the dynamics. Theoretical results [1, 9, 55] show that there exists a corresponding reverse-time SDE for the forward process. The forward process is represented as

$$\mathrm{d}\boldsymbol{X}_t = h(\boldsymbol{X}_t, t)\,\mathrm{d}t + g(\boldsymbol{X}_t, t)\,\mathrm{d}\boldsymbol{W}_t, \tag{3}$$

where $h(\cdot, t) : \mathbb{R}^d \to \mathbb{R}^d$ is the *drift* function, $g(\cdot, t) : \mathbb{R}^d \to \mathbb{R}^{d \times d}$ is the *diffusion* function, and $\boldsymbol{W}_t$ denotes the standard Wiener process. We follow the Itô interpretation of SDEs throughout this paper. The corresponding reverse-time SDE for Eq. (3) is given by

$$\begin{aligned}
\mathrm{d}\boldsymbol{X}_t &= \left(h(\boldsymbol{X}_t, t) - \nabla \cdot [g(\boldsymbol{X}_t, t)g(\boldsymbol{X}_t, t)^\top] - g(\boldsymbol{X}_t, t)g(\boldsymbol{X}_t, t)^\top \nabla \log f_{\boldsymbol{X}}(\boldsymbol{X}_t, t)\right)\mathrm{d}t \\
&\quad + g(\boldsymbol{X}_t, t)\mathrm{d}\bar{\boldsymbol{W}}_t,
\end{aligned} \tag{4}$$

where $\mathrm{d}\bar{\boldsymbol{W}}_t$ is the standard Brownian motion and $\nabla \cdot F(\boldsymbol{x}) := (\nabla \cdot f^1(\boldsymbol{x}), \nabla \cdot f^2(\boldsymbol{x}), \cdots, \nabla \cdot f^d(\boldsymbol{x}))^\top$ is the row-wise divergence of the matrix-valued function $F(\boldsymbol{x}) := (f^1(\boldsymbol{x}), f^2(\boldsymbol{x}), \cdots, f^d(\boldsymbol{x}))^\top \in \mathbb{R}^{d \times d}$. The issue with generating new samples from Eq. (4) is that we usually do not have access to the score function $\nabla \log f_{\boldsymbol{X}}(\boldsymbol{X}_t, t)$ and this quantity is approximated using a neural network $s_{\boldsymbol{\theta}} : \mathbb{R}^d \times [0, 1] \to \mathbb{R}^d$, which is trained by optimizing the denoising score-matching loss [55]

$$\mathcal{L}(\boldsymbol{\theta}) = \mathop{\mathbb{E}}_{t \sim \mathcal{U}[0,1]} \left[ \mathop{\mathbb{E}}_{\substack{\boldsymbol{X}_0 \sim p_{\boldsymbol{X}_0} \\ \boldsymbol{X}_t \sim p_{\boldsymbol{X}_t | \boldsymbol{X}_0}}} \left[ \lambda(t) \left\| s_{\boldsymbol{\theta}}(\boldsymbol{X}_t, t) - \nabla \log p_{\boldsymbol{X}_t | \boldsymbol{X}_0}(\boldsymbol{X}_t | \boldsymbol{X}_0) \right\|_2^2 \right] \right], \qquad (5)$$

where $\nabla \log p_{\boldsymbol{X}_t | \boldsymbol{X}_0}(\boldsymbol{X}_t | \boldsymbol{X}_0)$ is determined by the forward SDE (Eq. (3) [46]) and $\lambda(t)$ is designed to stabilise training.

# 4  Geometric Brownian Motion

Brownian motion, originally introduced to model random particle motion [16], is widely used in physics, biology, and signal processing to describe processes with independent and identically distributed (i.i.d.) increments. The resulting distribution is Gaussian following the Central Limit Theorem. For example, the Ornstein-Uhlenbeck SDE (OU-SDE) [13] models the position $Y_t$ of a Brownian particle as $\mathrm{d}Y_t = \mu \, \mathrm{d}t + \sigma \, \mathrm{d}W_t$, where $W_t$ is a Wiener process, yielding $Y_t = Y_0 + \mu t + \sigma W_t$, a Gaussian process with mean $\mu$ and variance $\sigma^2$. Alternatively, when the relative increments (or ratios) follow the Brownian motion, the resulting stochastic process is called the Geometric Brownian Motion (GBM). Black and Scholes [6] pioneered the use of GBM for modeling the evolution of stock prices and financial assets in mathematical finance. Just as the normal distribution plays a crucial role in Brownian motion, the log-normal distribution plays a vital role in the analysis of GBM. Formally, a random process $X_t$ is said to follow a Geometric Brownian Motion if it satisfies the SDE:

$$\mathrm{d}X_t = \mu X_t \, \mathrm{d}t + \sigma X_t \, \mathrm{d}W_t, \qquad (6)$$

where $W_t$ is the Wiener process, and $\mu$ and $\sigma$ are known as the *percentage drift* representing a general trend and *volatility coefficients* representing the inherent stochasticity, respectively. The solution of Eq. (6) $X_t$ evolves to follow a log-normal distribution with parameters $\mu$ and $\sigma^2$, i.e.,

$$X_t = X_0 \exp\left( \left( \mu - \frac{1}{2}\sigma^2 \right) t + \sigma W_t \right).$$

There exist several multivariate extensions of GBM [23]. We consider the element-wise extension of Eq. (6) for image data with the forward SDE for time $t \in [0, 1]$:

$$\mathrm{d}\boldsymbol{X}_t = \boldsymbol{\mu} \circ \boldsymbol{X}_t \, \mathrm{d}t + \sigma \boldsymbol{X}_t \circ \mathrm{d}\boldsymbol{W}_t, \qquad (7)$$

where $\circ$ denotes element-wise multiplication, $\boldsymbol{\mu} \in \mathbb{R}^d$, $\sigma > 0$ and $\boldsymbol{W}_t$ denotes the multivariate Wiener process. This can be written equivalently, using Itô's lemma, as

$$\mathrm{d}\log \boldsymbol{X}_t = \left( \boldsymbol{\mu} - \frac{\sigma^2}{2}\mathbf{1} \right) \, \mathrm{d}t + \mathrm{d}\boldsymbol{W}_t, \qquad (8)$$

where $\log$ is applied element-wise. The distribution of $\boldsymbol{X}_t$, as it evolves according to Eq. (8), has i.i.d. entries that are log-normally distributed with parameters $\boldsymbol{\mu}$ and $\sigma^2 \mathbb{I}$, $\mathbb{I}$ being the $d \times d$ identity matrix. Starting from a sample $\boldsymbol{X}_0$ from the unknown density $p_{\boldsymbol{X}_0}$, the solution to Eq. (8) is

$$\boldsymbol{X}_t = \boldsymbol{X}_0 \circ \exp\left( \left( \boldsymbol{\mu} - \frac{\sigma^2}{2}\mathbf{1} \right) t + \sigma \boldsymbol{W}_t \right).$$

This closed-form expression allows us to easily generate samples from the forward process at arbitrary time instants $t \in [0, 1]$. The samples at the end of the forward process are log-normally distributed. We now seek to derive the corresponding reverse-time SDE that would enable us to generate samples from the unknown density $p_{\boldsymbol{X}_0}$ starting from samples from the log-normal density. While one could use Eq. (4) to derive the corresponding reverse-time SDE, we propose a simpler approach by defining an auxiliary stochastic process $\boldsymbol{Y}_t = \log \boldsymbol{X}_t$ and leveraging *score change-of-variables formula* [44]. This allows us to rewrite Eq. (8) as

$$\mathrm{d}\boldsymbol{Y}_t = \left( \boldsymbol{\mu} - \frac{\sigma^2}{2}\mathbf{1} \right) \, \mathrm{d}t + \mathrm{d}\boldsymbol{W}_t. \qquad (9)$$

Forward SDE: $\mathrm{d}\boldsymbol{X}_t = \boldsymbol{\mu} \circ \boldsymbol{X}_t \, \mathrm{d}t + \sigma \boldsymbol{X}_t \circ \mathrm{d}\boldsymbol{W}_t$

$$\boldsymbol{X}_{k+1} = \boldsymbol{X}_k \circ \exp\left(\delta\left(\boldsymbol{\mu} - \frac{\sigma^2}{2}\mathbf{1}\right) + \sigma\sqrt{\delta}\boldsymbol{Z}_k\right)$$

$$\boldsymbol{X}_{k-1} = \boldsymbol{X}_k \circ \exp\left(-\delta\left(\boldsymbol{\mu} - \frac{3\sigma^2}{2}\mathbf{1}\right) + \delta\sigma^2\boldsymbol{X}_k \circ \nabla \log p_{\boldsymbol{X}_k}(\boldsymbol{X}_k, k) + \sqrt{\delta}\sigma\boldsymbol{Z}_k\right)$$

Reverse-Time SDE: $\mathrm{d}\boldsymbol{X}_t = \boldsymbol{X}_t \circ \exp\left(-(\boldsymbol{\mu} - \sigma^2\mathbf{1}) + \sigma^2\boldsymbol{X}_t \circ \nabla \log p_{\boldsymbol{X}_t}(\boldsymbol{X}_t, t)\right) \mathrm{d}t + \sigma\boldsymbol{X}_t \circ \mathrm{d}\boldsymbol{W}_t$

Figure 1: The forward and reverse-time SDEs for Geometric Brownian Motion (GBM). The forward SDE describes the evolution of a clean image sample to a noisy one that eventually becomes log-normally distributed, while the reverse-time SDE captures the dynamics of the process and generates new samples from the unknown density starting from log-normal noise. This is enabled by the knowledge of the unknown density manifesting through the score function.

The reverse-time SDE corresponding to the forward SDE in Eq. (9) can be obtained by invoking Eq. (4) and is given by

$$\mathrm{d}\boldsymbol{Y}_t = \left(\boldsymbol{\mu} - \frac{\sigma^2}{2}\mathbf{1} - \sigma^2 \nabla \log p_{\boldsymbol{Y}_t}(\boldsymbol{Y}_t, t)\right) \mathrm{d}t + \sigma\mathrm{d}\boldsymbol{W}_t, \tag{10}$$

where $\nabla \log p_{\boldsymbol{Y}}(\boldsymbol{Y}_t, t)$ is the score function corresponding to $\boldsymbol{Y}_t$ and $\mathbf{1}$ is a vector of all ones. We invoke the score change-of-variables formula [44] that allows us to represent $\nabla \log p_{\boldsymbol{Y}_t}(\boldsymbol{Y}_t, t)$ in terms of $\nabla \log p_{\boldsymbol{X}_t}(\boldsymbol{X}_t, t)$ as $\nabla \log p_{\boldsymbol{Y}_t}(\boldsymbol{Y}_t, t) = \mathbf{1} + \boldsymbol{X}_t \circ \nabla \log p_{\boldsymbol{X}_t}(\boldsymbol{X}_t, t)$. Thus, we rewrite Eq. (10) in terms of $\boldsymbol{X}_t$ and simplify it to obtain

$$\mathrm{d}\log\boldsymbol{X}_t = \left(\boldsymbol{\mu} - \frac{3\sigma^2}{2}\mathbf{1} - \sigma^2\boldsymbol{X}_t \circ \nabla \log p_{\boldsymbol{X}_t}(\boldsymbol{X}_t, t)\right) \mathrm{d}t + \sigma\mathrm{d}\boldsymbol{W}_t. \tag{11}$$

To simulate the reverse-time SDE on a computer, it must be discretized in time. We chose the time range $[0, 1]$ with $N$ steps, which results in a step-size of $\delta = \frac{1}{N}$ and for brevity, denote $\boldsymbol{X}_{k\delta}$ as $\boldsymbol{X}_k$, for $k = 0, \ldots, N-1$. In particular, we choose the Euler-Maruyama discretization scheme [20] for Eq. (11) to get

$$\log\boldsymbol{X}_{k-1} = \log\boldsymbol{X}_k - \delta\left(\boldsymbol{\mu} - \frac{3\sigma^2}{2}\mathbf{1} - \sigma^2\left(\boldsymbol{X}_t \circ \nabla \log p_{\boldsymbol{X}_t}(\boldsymbol{X}_t, t)\right)\big|_{t=k\delta}\right) + \sqrt{\delta}\sigma\boldsymbol{Z}_k, \tag{12}$$

where $\boldsymbol{Z}_k \sim \mathcal{N}(\mathbf{0}, \mathbb{I})$ (the standard normal distribution), and since the $\log$ operates element-wise, exponentiating both sides gives

$$\boldsymbol{X}_{k-1} = \boldsymbol{X}_k \circ \exp\left(-\delta\left(\boldsymbol{\mu} - \frac{3\sigma^2}{2}\mathbf{1}\right) + \delta\sigma^2\boldsymbol{X}_k \circ \nabla \log p_{\boldsymbol{X}_k}(\boldsymbol{X}_k, k) + \sqrt{\delta}\sigma\boldsymbol{Z}_k\right). \tag{13}$$

The update rule in Eq. (13) is similar to the EGD update rule in Eq. (2). Consider the optimization problem with a modification of the task error as

$$\boldsymbol{X}_{t+1} = \arg\min_{\boldsymbol{X}}\left[\bar{\ell}(\xi(\boldsymbol{X})) + \frac{1}{\eta}D_\phi(\boldsymbol{X}, \boldsymbol{X}_t)\right], \tag{14}$$

with the choice of $\xi : \mathbb{R}^d \to \mathbb{R}^d$ as $\xi^{(i)}(\boldsymbol{X}) = 0.5 \left(X^{(i)}\right)^2$ for $i = 1, 2, \cdots, d$. This leads to the following multiplicative update rule

$$\boldsymbol{X}_{k+1} = \boldsymbol{X}_k \circ \exp\left(-\eta \boldsymbol{X}_k \circ \nabla_{\boldsymbol{X}} \ell(\boldsymbol{X}) \mid_{\boldsymbol{X}=\boldsymbol{X}_k}\right). \tag{15}$$

Interestingly, if we assume that the density $p_{\boldsymbol{X}_k}(\boldsymbol{X}_k, k)$ is of the form $p_{\boldsymbol{X}_k}(\boldsymbol{X}_k, k) = \frac{1}{Z} \exp\left(-\ell(\boldsymbol{X}_k)\right)$, with $\eta = \delta \sigma^2$ and $\boldsymbol{\mu} = \frac{3\sigma^2}{2}$, then the corresponding sampling step in Eq. (13) is of the form

$$\boldsymbol{X}_{k-1} = \boldsymbol{X}_k \circ \exp\left(-\eta \boldsymbol{X}_k \circ \nabla_{\boldsymbol{X}} \ell(\boldsymbol{X}) \mid_{\boldsymbol{X}=\boldsymbol{X}_k} + \sqrt{\eta} \boldsymbol{Z}_k\right), \tag{16}$$

where $\boldsymbol{Z}_k \sim \mathcal{N}(\boldsymbol{0}, \mathbb{I})$. Therefore, the proposed sampler is structurally equivalent to the modified exponential gradient descent step in Eq. (15).

## 5   Multiplicative Score Matching

Following the definitions of explicit score-matching (ESM) loss and denoising score-matching (DSM) loss for the additive noise case [56], we propose the multiplicative counterparts $\mathcal{L}_{\text{M-ESM}}(\boldsymbol{\theta})$ and $\mathcal{L}_{\text{M-DSM}}(\boldsymbol{\theta})$ as follows:

$$\mathcal{L}_{\text{M-ESM}}(\boldsymbol{\theta}) = \mathop{\mathbb{E}}_{\boldsymbol{X}_t \sim p_{\boldsymbol{X}_t}} \left[\frac{1}{2} \|\boldsymbol{X}_t \circ \nabla \log p_{\boldsymbol{X}_t}(\boldsymbol{X}_t) - \boldsymbol{X}_t \circ s_{\boldsymbol{\theta}}(\boldsymbol{X}_t, t)\|_2^2\right], \quad \text{and} \tag{17}$$

$$\mathcal{L}_{\text{M-DSM}}(\boldsymbol{\theta}) = \mathop{\mathbb{E}}_{\substack{\boldsymbol{X}_0 \sim p_{\boldsymbol{X}_0} \\ \boldsymbol{X}_t \sim p_{\boldsymbol{X}_t|\boldsymbol{X}_0}}} \left[\frac{1}{2} \|\boldsymbol{X}_t \circ \nabla \log p_{\boldsymbol{X}_t|\boldsymbol{X}_0}(\boldsymbol{X}_t|\boldsymbol{X}_0) - \boldsymbol{X}_t \circ s_{\boldsymbol{\theta}}(\boldsymbol{X}_t, t)\|_2^2\right]. \tag{18}$$

The two types of score-matching loss functions are related as follows.

**Theorem 5.1** (Multiplicative Denoising Score-Matching). *Under standard assumptions on the density and the score function [24, 53] over the positive orthant $\mathbb{R}^{+d}$, the multiplicative explicit score-matching (M-ESM) loss given in Eq. (17) and multiplicative denoising score-matching (M-DSM) loss given in Eq. (18) are equivalent up to a constant, i.e., $\mathcal{L}_{\text{M-DSM}}(\boldsymbol{\theta}) = \mathcal{L}_{\text{M-DSM}}(\boldsymbol{\theta}) + C$, where $C$ is independent of $\boldsymbol{\theta}$.*

The proof is provided in the supplementary material. The usefulness of this result is explained next. We need the marginal score function $\nabla \log p_{\boldsymbol{X}_t}(\boldsymbol{X}_t)$ in the reverse-time SDE Eq. (13) but optimizing Eq. (17) is intractable since we do not have access to the "true" marginal score. The theorem provides us with a means to optimize for $s_{\boldsymbol{\theta}}$ in terms of the conditional score $\nabla \log p_{\boldsymbol{X}_t|\boldsymbol{X}_0}(\boldsymbol{X}_t|\boldsymbol{X}_0)$, which can be derived from the forward SDE. The challenge in leveraging Eq. (13) to generate new samples arises from our lack of knowledge of $\nabla \log p_{\boldsymbol{X}_t}(\boldsymbol{X}_t)$. This function must be estimated by some form of score-matching. To this end, we propose the following score-matching loss

$$\mathcal{L}_{\text{M-DSM}}(\boldsymbol{\theta}) = \mathop{\mathbb{E}}_{\substack{\boldsymbol{X}_0 \sim p_{\boldsymbol{X}_0} \\ \boldsymbol{X}_t \sim p_{\boldsymbol{X}_t|\boldsymbol{X}_0}}} \left[\frac{1}{2} \|\boldsymbol{X}_t \circ \nabla \log p_{\boldsymbol{X}_t|\boldsymbol{X}_0}(\boldsymbol{X}_t \mid \boldsymbol{X}_0) - \boldsymbol{X}_t \circ s_{\boldsymbol{\theta}}(\boldsymbol{X}_t, t)\|_2^2\right]. \tag{19}$$

In practice, this choice of the loss function allows us to train the score network $s_{\boldsymbol{\theta}}$ using samples from the forward SDE in Eq. (7) and the corresponding conditional score $\nabla \log p_{\boldsymbol{X}_t|\boldsymbol{X}_0}(\boldsymbol{X}_t|\boldsymbol{X}_0)$ evaluated at discrete instants of time $t = k\delta$ can be computed using the forward SDE and the expression for the target in the loss function is given by

$$\boldsymbol{X}_t \circ \nabla \log p_{\boldsymbol{X}_t|\boldsymbol{X}_0}(\boldsymbol{X}_t|\boldsymbol{X}_0) = -\left(\boldsymbol{1} + \frac{1}{\sigma^2 t\delta} \left(\log \boldsymbol{X}_k - \log \boldsymbol{X}_0 - t\delta \left(\boldsymbol{\mu} - \frac{\sigma^2}{2}\boldsymbol{1}\right)\right)\right). \tag{20}$$

The proposed loss function in Eq. (19) is the multiplicative noise counterpart of the denoising score-matching loss proposed by Song et al. [55] for additive noise. To the best of our knowledge, this formulation of the score-matching loss and its manifestation in the multiplicative noise setting is new. It would be appropriate to remark here that the score term in Eq. (13) also arises in the score-matching loss proposed by Hyvärinen [25] for non-negative real data given by

$$\mathcal{L}_{\text{NN}}(\boldsymbol{\theta}) = \frac{1}{2} \mathop{\mathbb{E}}_{\boldsymbol{X}_0 \sim p_{\boldsymbol{X}_0}} \left[\|\boldsymbol{X}_0 \circ \nabla \log p_{\boldsymbol{X}_0}(\boldsymbol{X}_0) - \boldsymbol{X}_0 \circ s_{\boldsymbol{\theta}}(\boldsymbol{X}_0)\|_2^2\right], \tag{21}$$

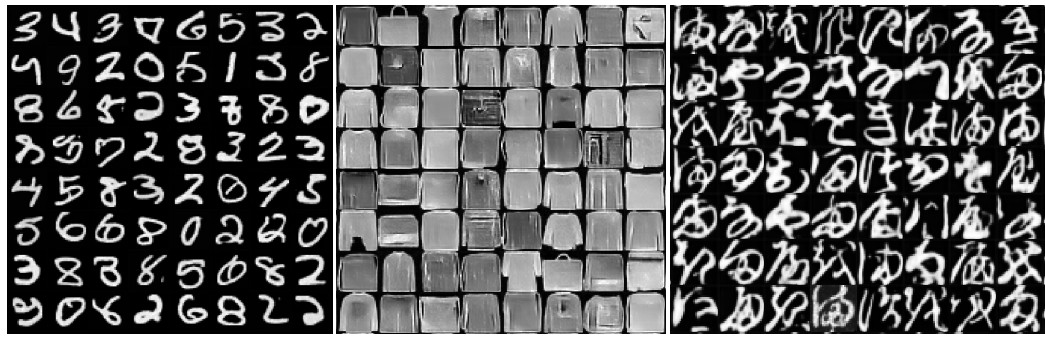

Figure 2: Uncurated sample images generated from MNIST, Fashion-MNIST and Kuzushiji MNIST datasets, corresponding to the score model with minimum score-matching loss during training.

where $\nabla \log p_{\boldsymbol{X}_0}(\boldsymbol{X}_0)$ is the true score. Hyvärinen [25]'s formulation is static in the sense that it does not leverage the SDE, whereas we do. Hyvärinen [25]'s score-matching loss can also be seen as an instance of the multiplicative explicit score-matching loss (M-ESM) for $t = 0$. Hyvärinen [25]'s motivation for introducing this loss function is to avoid the singularity at the origin for non-negative data. Our framework encapsulates this variant of the score-matching loss as a special case. This is primarily due to the structure of GBM that assumes the log-normal distribution which implicitly restricts the samples to be positive. Thus, our framework generalizes the score-matching loss proposed by Hyvärinen [25] to the case of multiplicative noise.

---

**Algorithm 1** Multiplicative updates for generation using Geometric Brownian Motion (GBM).

---
**Require:** $\sigma, \delta, \boldsymbol{\mu}$, trained score network $s_{\boldsymbol{\theta}}$
1: **for** $k \leftarrow 0$ to $N - 1$ **do**
2: $\quad \boldsymbol{Z}_k \sim \mathcal{N}(\mathbf{0}, \mathbb{I})$
3: $\quad \boldsymbol{X}_{k-1} = \boldsymbol{X}_k \circ \exp\left(-\delta\left(\boldsymbol{\mu} - \frac{\sigma^2}{2}\mathbf{1}\right) + \delta\sigma^2 \boldsymbol{X}_k \circ s_{\boldsymbol{\theta}}(\boldsymbol{X}_k, k) + \sigma\sqrt{\delta}\boldsymbol{Z}_k\right)$
4: **end for**

---

## 5.1 Image Generation using Multiplicative Score Matching

The goal in diffusion-based image generative modeling is to construct two stochastic processes, as illustrated in Fig. 1 – the forward process to generate a noisy version of a clean image and the reverse process to enable us to sample from the unknown density $p_{\boldsymbol{X}_0}$. For the forward model, starting from an image $\boldsymbol{X}_0$ coming from the unknown density, the forward SDE in Eq. (8) can be used to generate noisy versions of $\boldsymbol{X}_0$ as follows

$$\boldsymbol{X}_{k+1} = \boldsymbol{X}_k \circ \exp\left(\delta\left(\boldsymbol{\mu} - \frac{\sigma^2}{2}\mathbf{1}\right) + \sqrt{\delta}\sigma\boldsymbol{Z}_k\right), \tag{22}$$

for $k = 0, \ldots, N - 2$, and $\boldsymbol{X}_{N-1}$ is log-normally distributed and $\boldsymbol{Z}_k \sim \mathcal{N}(\mathbf{0}, \mathbb{I})$. For the reverse process, i.e., generation, we can generate samples from the reverse-time SDE in Eq. (11) using the discretized version of the reverse-time SDE in Eq. (13) and the score model $s_{\boldsymbol{\theta}}(\cdot)$ trained with the loss defined in Eq. (19) in place of the true score function $\nabla \log p_{\boldsymbol{X}_t}(\cdot)$. The new generation/sampling procedure is summarized in Algorithm 1. The algorithm takes as input the parameters $\sigma, \delta, \boldsymbol{\mu}$ and the trained score network $s_{\boldsymbol{\theta}}$ and generates samples from the unknown density $p_{\boldsymbol{X}_0}$ by iterating over $N$ steps. The algorithm starts with a sample $\boldsymbol{X}_{N-1}$ from the log-normal distribution and iteratively updates the sample using the reverse-time SDE in Eq. (13). The final output should be a sample $\boldsymbol{X}_0 \sim p_{\boldsymbol{X}_0}$.

# 6  Experiments

We evaluate the generative performance of the proposed model[1] by training the score model on standard datasets such as MNIST, Fashion-MNIST and Kuzushiji MNIST dataset used by Cornford et al. [11]. The datasets are split as $60,000$ images for training and $10,000$ images for testing. All images are rescaled to have pixel values in the range $[1, 2]$. Note that the proposed framework requires a non-negative dynamic range of pixel values. We choose $N = 1000$ discretization levels for the forward SDE (7) and leads to the step size $\delta = 1/N$. During sampling, we observed that the same step-size did not always work and we had to work with smaller step-sizes for each of the three datasets. The model is trained using the M-DSM loss defined in Eq. (19). The hyperparameters $\mu = \frac{\sigma^2}{2}\mathbf{1}$, $\sigma$ and $\delta$ are set to 0.8 and 0.001, respectively. The model is trained for 200000 iterations and the checkpoints are saved every 5000 iterations as mentioned in [52] on two NVIDIA RTX 4090 and two A6000 GPUs. We perform exponential moving average for the saved checkpoints every 50000 iterations. The generated samples are shown in Figure 2, from where we observe that the visual quality of the generated images matches is on par with that of the ground truth. For quantitative assessment, we use Fréchet Inception Distance (FID) [19] and Kernel Inception Distance (KID) [5] measured between $10,000$ images from the test dataset and the same number of generated images. Lower FID and KID values indicate superior generative performance. While both these metrics are not commonly used to quantify the generative performance for grayscale images, we follow Xu et al. [61] and report these numbers for transparency and reproducibility (cf. Supplementary Material).

# 7  Conclusions

We proposed a novel generative model based on Geometric Brownian Motion (GBM) and a new technique for score-matching. We showed that the GBM framework is a natural setting for modeling non-negative data and that the new multiplicative score-matching loss can be used effectively to train the model. The model is capable of generating new samples from image datasets like MNIST, Fashion MNIST and Kuzushiji MNIST. The results are promising from a generative modeling perspective. The multiplicative score matching framework can also be suitably adapted for image denoising and restoration tasks where the forward model has multiplicative noise as opposed to the widely assumed additive noise. While this work focuses on log-normal noise, other distributions such as the gamma distribution, could also be considered with associated SDEs. This would broaden the applicability of the model to datasets and domains where various types of multiplicative noise are prevalent such as optical coherence tomography [33] and synthetic aperture radar [17], enabling more robust and versatile generative and restoration capabilities. Starting off with the results shown in the paper, one could also extend applicability of the proposed model to high-resolution images. Application to non-image data, such as financial time-series, is another potential direction for further research.

# Limitations

The proposed generative model requires a large amount of training data and computational resources to achieve good performance, which can be a constraint in some applications. In the true spirit of data-driven generation, some of the generated images do not have the same *semantic* meaning as samples from the source dataset. Incorporating semantics into generative modeling is a research direction by itself. Instead of cherry-picking the results, we reported them as obtained to highlight both the strengths and limitations of the proposed approach. The choice of hyperparameters, such as the noise schedule and learning rate, which are carefully tuned, can affect the performance of the model. However, this limitation is true of all deep generative models and not unique to ours.

**Broader Impact**: The proposed approach of leveraging the GBM and multiplicative score-matching is novel and has the potential to advance the field of generative modeling along new lines. The model may find natural applicability in financial time-series modeling, forecasting, and generation. Ethical concerns pertaining to the use of generative models and the potential for misuse by generating biased, fake, or misleading content are all pervasive and the proposed framework is no exception.

---

[1]Code for this paper is available at `https://anonymous.4open.science/r/gbm_dale-CC20`

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
