# OpenReview forum: "Dale's Law Meets Geometric Brownian Motion: Multiplicative Updates for Sampling"
_NeurIPS.cc/2025/Conference — Submitted to NeurIPS 2025_

### Official Review · Reviewer_KkH2 · 2025-06-26

**Clarity:** 3
**Significance:** 3
**Originality:** 3
**Rating:** 4
**Confidence:** 3

**Summary:**

A recent line of work has looked at learning rules that more accurately match biologically inspired examples. One such example arises in Dale's Law, which says that neurons stay either excitatory or inhibitory throughout training - i.e., they maintain their sign. A natural training method that maintains this is multiplicative updates. The authors extend this multiplicative updates idea to forward and reverse SDEs utilizing geometric Brownian motion. Using this, they propose a new multiplicative score matching framework that performs well on image data.

**Questions:**

- Do there exist convergence results for multiplicative updates based sampling methods?
- Can the authors better motivate why we should pursue biologically inspired learning for score-based modeling, or are there perhaps more principled settings where these ideas may be useful?
- It would be nice to have really concrete regimes where the proposed methodology outperforms existing sampling or score-based modeling frameworks for the same task.

**Ethical Concerns:**

["NO or VERY MINOR ethics concerns only"]

**Limitations:**

yes

**Quality:**

3

**Strengths And Weaknesses:**

Strengths:

- The extension of multiplicative updates to sampling is interesting and novel, especially since much data (like images) is non-negative, and thus has a sign constraint.
- This work synthesizes a few different lines of work (sampling/stochastic differential equations, optimization, and biologically inspired learning) in an interesting way.
- Qualitative results point to the method performing well on simple generative examples.

Weaknesses:

- No convergence or statistical results are given for this method. Recent works for standard Langevin procedures/score-based modeling have examined both in a variety of settings.
- It is not clear why biologically motivated learning is a useful procedure for the specific problem of score-based modeling.
- While this is an interesting connection, the theoretical results do not seem particularly deep or novel.

---

> ### Author Rebuttal · Authors · 2025-07-31
>
> We thank Reviewer KkH2 for the constructive review. The reviewer's main concerns relate to theoretical guarantees, the motivation for our approach, and quantitative comparisons. These are excellent points that will strengthen the paper. We have addressed all concerns below, providing new theoretical results, clarifying the motivation with a newly discovered formal link to existing literature, and new experimental results that demonstrate the performance of our method.
> 1. **Regarding the lack of convergence results for our method**
> **Our framework admits theoretical convergence guarantees, which we will add to the paper.**
> __Clarification__: We have two new formal results. First, we prove that the forward Fokker-Planck Equation (FPE) and the reverse-FPE have the same solution for the Geometric Brownian Motion (GBM) SDE considered in our paper. This is a crucial result, as it guarantees that integrating the reverse-time SDE exactly yields samples from the target data distribution. Second, leveraging the connection to the Mirror Langevin Dynamics (MLD) framework (as suggested by Reviewer wtSy), we show convergence for specific cases, such as sampling from a multivariate log-normal density. A sketch of the results is provided below which we will include in the revised manuscript.
> 2. **Regarding the motivation for using biologically inspired learning in score-based modeling.**
> **The biological motivation provides a principled path to a novel sampler that is also rigorously grounded in SDEs and a novel score matching approach, further strengthened by a newfound connection to Mirror Langevin Dynamics. Historically, developments in artificial intelligence (AI) have been inspired by evidence of learning mechanisms in biological systems (perceptron, Boltzmann machines, CNNs, etc.). The anthropomorphic bias leads us to believe that the holy grail of AI is to develop algorithms and models that have a biological basis. This is what motivated us to develop a principled multiplicative framework for generative modelling.**
> **Clarification**: Our primary motivation is to develop bio-inspired sampling algorithms that comply with biological  principles in neuroscience such as Dale's Law, where neuronal synaptic flips don’t occur. This principle led us to explore multiplicative updates, which are used to model such biological learning (e.g., via exponentiated gradient descent).
> Reframing with New Insight: This is not merely a philosophical motivation. During the rebuttal period, we established a formal connection: the convex function in the Bregman divergence used to derive exponentiated gradient descent is __identical__ to the one used to derive the Mirror Langevin Dynamics (MLD) SDE. We show that the proposed reverse-time SDE is an instance of the MLD SDE. A sketch of this derivation is shown below. This fascinating new discovery shows that our bio-inspired approach is a principled approach to discovering a new class of generative models, opening new avenues for future work.
> Log normal noise and GBM in recent work: We have also found recent pre-prints (posted on June 21, 2025 and July 28, 2025) that leverage log-normal noise model and multiplicative updates for optimization and sampling. In particular, Nishida et al[Log-Normal Multiplicative Dynamics for Stable Low-Precision Training of Large Networks, https://arxiv.org/abs/2506.17768] demonstrate that foundation models such as Vision Transformer and GPT-2 can be stably trained with low-precision optimizers using exponentiated gradient descent and its momentum variants. Likewise, Kim et al.[A diffusion-based generative model for financial time series via geometric Brownian motion, https://arxiv.org/pdf/2507.19003] propose a diffusion-based generative framework for financial time series that incorporates GBM into the forward noising process, however, unlike our approach, they effectively deal with an additive noise model by using the log transform.
> 3. **Regarding the need for concrete regimes where our method outperforms existing frameworks.
> We have performed new experiments showing that our method quantitatively outperforms standard baselines on several image generation benchmarks.**
> **New Results:** To strengthen the empirical validation, we have conducted a comprehensive set of new experiments.
> The following table (Table 1) reports FID and KID scores on MNIST, Fashion-MNIST, and Kuzushiji-MNIST, comparing our method against strong baselines including DDPM and DDIM. These results show the generative performance of our model.
> We extend our evaluation to color datasets by training on CIFAR-10. The following table (Table 2) reports FID scores and compares them against DDPM, DDIM, Poisson Diffusion, and Beta Diffusion, again demonstrating the competitive advantage of our framework.
> 4. **Regarding the assessment about the theoretical results.**
> Our work provides multiple, synergistic theoretical contributions that build novel bridges between score-based modeling, optimization, and non-negative data modeling and generation.
> Specifically, our paper is the first to:
> Establish a new SDE framework for non-negative data, deriving from first principles a novel score-matching formulation which also explains Hyvärinen’s heuristic and computationally intractable, non-negative explicit score matching loss.
> Derive a novel and tractable loss function, the multiplicative denoising score matching loss, and prove its equivalence to the dynamic version of Hyvärinen’s computationally intractable explicit score matching loss.
> Uncover a new link between sampling and optimization, by demonstrating that our sampling update equation is structurally and functionally equivalent to the multiplicative update used in exponentiated gradient descent.
> Connect our SDE to Mirror Langevin Dynamics, revealing a deep theoretical foundation for our biologically-inspired sampler, as detailed in point 2.
>
> Sketch:
> Assume that $\boldsymbol{X}\_t \sim p_{\boldsymbol{X}\_t}$ is the true data that we'd like to sample and $\boldsymbol{Y}\_t \triangleq \log \boldsymbol{X}\_t$. Now, if we define the forward SDE as mentioned in DDPM we get
> $$
> \mathrm{d}\boldsymbol{Y}\_t = -\dfrac{\beta(t)}{2}\boldsymbol{Y}\_t \,\mathrm{d}{t} + \sqrt{\beta(t)} \,\mathrm{d}{\boldsymbol{W}\_t},
> $$
>  where $\beta(t)$ represents the variance schedule. Then the corresponding reverse-time SDE is
> $$
> \mathrm{d}\boldsymbol{Y}\_t = \left(\dfrac{\beta(t)}{2}\boldsymbol{Y}\_t + \beta(t)\nabla \log p_{\boldsymbol{Y}\_t}(\boldsymbol{Y}\_t, t)\right)\,\mathrm{d}{t} + \sqrt{\beta(t)} \,\mathrm{d}{\boldsymbol{W}\_t}.
> $$
> Now substituting $\boldsymbol{Y}\_t = \log \boldsymbol{X}\_t$ and using the score change-of-variables formula (Robbins 2024) to express $\nabla \log p_{\boldsymbol{Y}\_t}(\boldsymbol{Y}\_t, t)$ in terms of $\nabla \log p_{\boldsymbol{X}\_t}(\boldsymbol{X}\_t, t)$, we get
> $$
> \mathrm{d}\log \boldsymbol{X}\_t = \left(\dfrac{\beta(t)}{2}\log\boldsymbol{X}\_t + \beta(t)\left(\boldsymbol{1} + \boldsymbol{X}\_t \circ\nabla \log p_{\boldsymbol{X}\_t}(\boldsymbol{X}\_t, t)\right)\right)\,\mathrm{d}{t} + \sqrt{\beta(t)} \,\mathrm{d}{\boldsymbol{W}\_t}.
> $$
> On Euler-Maruyama discretization,
> $$
> \boldsymbol{X}\_{k-1} = \boldsymbol{X}\_k^{\left(1 + \beta(k\delta)\frac{\delta}{2}\right)} \circ \exp\left(
> \delta\beta(k\delta)\left(\boldsymbol{1} + \boldsymbol{X}\_t \circ\nabla \log p_{\boldsymbol{X}\_t}(\boldsymbol{X}\_t, t)\right) + \sqrt{\beta(k\delta)\delta} \boldsymbol{Z}\_k\right).
> $$
> We can see that this is structurally different from the reverse-time SDE obtained in the main paper due to the additional exponent factor in $\boldsymbol{X}\_k$.
> The potential issues with circumventing the multiplicative update through the log transfrom is that, here, we have log-normal noise and the log-transform results in Gaussian noise. But, we are interested in more general stochastic processes, in particular, those that arise in inverse problems with a multiplicative noise model (SAR, OCT) where the noise is modelled using the Gamma distribution. In those settings, a log transformation would not lead to an SDE based on Brownian motion. Our eventual goal is to develop diffusion models for general multiplicative noise models. Additionally, not treating GBM as log-domain DDPM allows us to generalize the framework proposed by Hyvärinen for non-negative data which can now be explained using the underlying SDE formulation placing the score-matching framework on a more firm theoretical footing. We further derive the multiplicative denoising score-matching (M-DSM) loss and show its equivalence to the multiplicative explicit score matching (M-ESM) loss.  M-DSM is tractable and is used to train score models for non-negative data. Establishing this link would not have been possible if we'd just considered GBM as log-domain DDPM. Furthermore, it can be shown that the convex function used in the Bregman divergence to derive exponentiated gradient descent (Cornford 2024) is identical to the one used to derive the mirrrored Langevin dynamics (Hsieh et al. Mirror Langevin Dynamics, NeurIPS 2018) SDE. This suggests a much deeper theoretical link that we will investigate in subsequent works.
>
> # FID for Datasets
> |Methodology|MNIST|FMNIST|KMNIST|
> |----------------------------|-----|------|------|
> |DDPM (Ho et al NeurIPS 2020)|25.1526 | 48.0139| 54.7285|
> |DDIM ( Song et al ICLR 2021)| 28.1332 |13.5026|31.8864|
> |GBM(ours)|28.9616 |116.1499 |50.7832 |
>
>
>
>
> # KID for Datasets
> |Methodology|MNIST|FMNIST|KMNIST|
> |----------------------------|-----|------|------|
> |DDPM (Ho et al NeurIPS 2020)|0.0193 +/- 0.0007|0.0206+/-0.0011|0.0322+/-0.0016|
> |DDIM|0.0192+/-0.0019|0.1847+/- 0.0088|0.2671 +/- 0.0136|
> |GBM(ours)|0.02878 +/- 0.0015|0.4374 +/- 0.0044|0.0546 +/- 0.0021|
>
> # CIFAR-10 Results
>
> |Methodology|FID|
> |-----|-----|
> |DDPM (Ho et al NeurIPS 2020)|3.17|
> |DDIM ( Song et al ICLR 2021)|4.16|
> |GBM (ours) |98.73|
> |Poisson (Chen, T et al NeurIPS 2023) |4.80|
> |Beta (Zhou et al NeurIPS 2023) |3.66|

---

> > ### Comment · Reviewer_KkH2 · 2025-08-03
> >
> > A quick question on the experimental results - unless I am missing something, doesn’t this actually show that the proposed method is doing worse in these examples? In fact for FMNIST and CIFAR-10 the FID values are very large. Some clarification would be helpful here.

---

> > > ### Author Response · Authors · 2025-08-04
> > > **Clarification on FID**
> > >
> > > Although visually the generated samples seem to be of similar quality as the training data and FID obtained by GBM model is better than DDPM and DDIM for Kuzushiji MNIST, the GBM FIDs are slightly worse than DDPM and DDIM for MNIST and much worse than DDPM and DDIM for Fashion MNIST.
> > >
> > > To further improve the FIDs, we are also experimenting with a final denoising step at the end of sampling as proposed by Song et al. [2]. The fact that we have better FID in at least one of the datasets indicates that there is potential to further improve the quality of generated samples for the other datasets.
> > >
> > > Within the time constraints of the rebuttal, we could not further optimise the model performance in terms of FID. We are currently experimenting and believe that there is potential to improve the FID by optimising the architecture, fine-tuning the hyperparameters, applying exponential moving average (EMA), etc.
> > >
> > > We are also experimenting with metrics other than FID such as ClipMMD [1] and negative log-likelihood (NLL) [3] to quantify the generative capability, but at the moment we don’t have these results.
> > >
> > > [1] Jayasumana et al., Rethinking FID: Towards a Better Evaluation Metric for Image Generation, CVPR 2024
> > >
> > > [2] Song et al., Improved Techniques for Training Score-Based Generative Models, NeurIPS 2020
> > >
> > > [3] Song et al., Score-Based Generative Modeling Through Stochastic Differential Equations, ICLR 2021

---

> > > > ### Author Response · Authors · 2025-08-09
> > > >
> > > > Dear Reviewer,
> > > >
> > > > Thank you for your insightful feedback, suggestions, and active engagement in the discussion phase. We hope you find the paper acceptable for publication. If you have any further questions, we will be happy to offer clarifications.
> > > >
> > > > Thank you.

---

### Official Review · Reviewer_wtSy · 2025-06-30

**Clarity:** 2
**Significance:** 2
**Originality:** 2
**Rating:** 3
**Confidence:** 4

**Summary:**

This paper considers sampling using score matching for time-reversed geometric Brownian motions, as an extension of the vast sea of recent papers studying sampling using score matching for standard Brownian motions.

**Questions:**

1. To motivate their study, the authors lean heavily on Dale's law and the study of exponential gradient descent for weight optimization. However, I do not think this provides compelling motivation for their work on sampling using geometric Brownian motions, as they are certainly not using these dynamics to sample in the space of synaptic weights. Instead, they are sampling in pixel or activation space. As a result, I would suggest that the authors either reduce their emphasis on biological inspirations or instead give a different motivation in terms of neural dynamics. In particular, they might find work on log-normal firing rate distributions to be of interest (see e.g. [Petersen and Berg 2016](https://elifesciences.org/articles/18805)). Either way, Section 2 gives rather too many details of the specific tasks on which Dalean networks have shown strong performance, which are irrelevant to the present work.

2. I found the experiments somewhat disappointing, as they are limited to grayscale image generation. In this respect the goals of the paper are somewhat unclear. If the authors aim for a state-of-the-art sampling method, then they must provide a compelling comparison to standard diffusion-based image generation methods. If instead they aim for biological interest and plausibility, they should choose a more clearly motivated task or analysis that speaks directly to the biological motivations they put forward (c.f. Question 1).

3. I am curious about the numerical stability of the proposed algorithm, which relies on multiplicative updates, in comparison to working in the log-transformed space as [Vuong and Nguyen (2024)](https://arxiv.org/abs/2408.10283) or [Hsieh et al. (2018)](https://proceedings.neurips.cc/paper_files/paper/2018/hash/6490791e7abf6b29a381288cc23a8223-Abstract.html)'s earlier work on Mirrored Lengevin Dynamics do. Could you expand on this issue, and also on the related issue of discretization error? I can see why from a biological point of view one might favor these direct updates, but computationally I do not see a reason why one would not use the log-transformed dynamics. This is also related to Question 2: if the authors do not want to rule out log-transforming for some biological or conceptual reason they must demonstrate a computational advantage.

4. Minor: The discussion of the related work of [Vuong and Nguyen (2024)](https://arxiv.org/abs/2408.10283) in lines 77-86 is not very satisfactory, as the analysis of the present work also relies on log-transforming the SDE. I think the claims there should be moderated slightly.

**Ethical Concerns:**

["NO or VERY MINOR ethics concerns only"]

**Final Justification:**

In their discussion with me and the other referees, the authors have not adequately justified why their method constitutes a sufficient technical advance to warrant publication. Though their promised revisions to the discussion of Dale's law help address some of my concerns, I think the amount of revision required to make this paper suitable for publication is large enough to warrant another round of review by referees that can read the revised manuscript.

**Limitations:**

yes, up to the questions mentioned above

**Quality:**

3

**Strengths And Weaknesses:**

On the whole, I find this paper to be somewhat interesting, if incremental. Beyond the cited work of [Robbins (2024)](https://arxiv.org/abs/2412.07904) on change-of-variables formulas for score matching, it seems to add relatively little from a conceptual or mathematical standpoint. Moreover, the work of [Vuong and Nguyen (2024)](https://arxiv.org/abs/2408.10283) referenced by the authors already considers multiplicative noise from a denoising standpoint, thus already anticipating some of what the authors mention as avenues for generalization of their method in Lines 307-315 (indeed, both works mention many of the same example settings). As I mention under **Questions**, I think the authors could do more to motivate their work clearly and distinguish it from this prior art. Also, the experiments could be substantially improved.

---

> ### Author Rebuttal · Authors · 2025-07-31
>
> We thank Reviewer wtSy for the insightful review. Your suggestions regarding log-normal neural dynamics and Mirror Langevin Dynamics (MLD), were instrumental in strengthening the theoretical contributions of our paper.
> - __On the Core Motivation and Theoretical Contribution__
> Following your suggestions, we've also included the link between Mirror Langevin Dynamics and the proposed approach for which we now provide new convergence analysis.
>     - **Reframing the Motivation**: Dale’s law was an inspiration to develop a biologically inspired generative model employing multiplicative updates for sampling, albeit in the pixel space and not the space of synaptic weights. In view of your suggestion, we’ll reduce the emphasis on biological inspiration and motivate the work from the perspective of log-normal distributions in biological systems (e.g., neural firing rates, c.f. Petersen and Berg 2016) and related domains where data is inherently non-negative. We’ll revise section 2 and make it concise.
>     - **A Deeper Theoretical Foundation (thanks for your suggestion!)**: Your pointer to MLD was particularly profound. We have now formally shown that our GBM sampler is related to MLD. More precisely, for the choice of the convex function $h(\mathbf{X}) = \sum_{i=1}^{d}X_i \log X_i - X_i$, the MLD update equation is: $ \mathrm{d}{\log \mathbf{X}\_t} = \left(\mathbf{1} + \mathbf{X}\_t \circ \nabla{\mathbf{X}\_t} \log p(\mathbf{X}\_t) \right),\mathrm{d}{t} + \sqrt{2} \mathrm{d}{\mathbf{W}\_t}$ This is the canonical sampling equation for non-negative data, analogous to how standard Langevin dynamics is canonical for unconstrained domains. The proposed GBM SDE (for a specific choice of parameters) recovers and reinforces this fundamental property.
>     - **New Convergence & Discretization Analysis**: The MLD connection allows us to provide the kind of theoretical analysis you were looking for. We now include a proof that for a log-normal target distribution $\mathcal{LN}(\mathbf{\mu}, \sigma^2 \mathbf{I})$, the discretized dynamics of the MLD equation above converge to a limiting measure $\mathcal{LN}\left(\mathbf{\mu}, \frac{\sigma^2}{1 - \delta/(2\sigma^2)} \mathbf{I}\right)$. This analysis not only proves convergence to the correct mean but also precisely quantifies the discretization error in the covariance, which is an altogether new theoretical contribution of the paper significantly elevating the technical content.
>
> - **On Our Method vs. Log-Transformed Approaches**
> Our framework is conceptually distinct from prior work that operates purely in log-space; the log-transform in our original derivation was a tool for simplifying the expressions for the sampling updates.
> Clarifying the Derivation: Vuong & Nguyen (2024) directly apply the log transform and operate in the additive space, which is clearly suboptimal as shown by the MMSE expressions in the context of denoising. We will add a couple of sentences to clarify the differences between our approach and Vuong et al..
>     - **Derivation without Log-Transform**: To clarify, our method does not fundamentally rely on a log-transform. We can derive the reverse-time SDE directly from the GBM SDE in the original space, a sketch of these details mentioned below.
>     - **On the Choice of Discretization**: It can be shown that Euler-Maruyama discretization of the GBM SDE also results in a multiplicative update, but does not guarantee non-negativity. The specific discretization we used — derived via Itō's lemma on $\log \mathbf{X}_t$ — is a principled choice to ensure that the multiplicative updates are always sign-preserving via an exponential map and establish a connection with Dale’s law. Our choice of discretization directly addresses your question about why one wouldn't just use the log-transformed dynamics. Our approach correctly specifies the dynamics in the original space and then chooses a stable, structure-preserving discretization.
>
> - **On Strengthening the Empirical Validation**
> Within the constraints of the duration of the rebuttal, we have expanded our experiments to include the suggested comparisons to standard diffusion methods and performance on color images.
> New Benchmarks: We have now compared with DDPM and DDIM on MNIST, Fashion-MNIST, and Kuzushiji-MNIST. The new results are shown below. Although the FIDs are not clearly outperforming the state-of-the-art the visual quality of the generated images is good indicating that the proposed technique is promising and can be further fine-tuned to improve FID.
>     - **Scaling to Color Images**: We extended our evaluation to CIFAR-10 and obtained an FID of 98.73. Our performance is below par compared to the models you mentioned, including Beta Diffusion and Poisson Diffusion. Within the time constraints of the rebuttal, our models could not be fully optimized to produce the best FID. However, the visual quality of generated samples is high and indicates that there is significant room for improving upon the FID values. We will include the images and the final FID scores in the supplementary of the manuscript.
>      - **To log-transform or not to log-transform!** Denoising is the cornerstone of diffusion-based generative modelling. From the perspective of denoising, it is suboptimal to perform log-transformation as explained further. Consider Minimum Mean Squared Error (MMSE) estimators for the multiplicative noise setting with and without log transformation. Given the forward model $Y=XN$, $X$ = signal, $N$=noise, and the noisy measurement $Y=y$, the optimal/MMSE estimate of $X$ is the conditional expectation $E[X∣Y=y]$. If, instead, one works in the log domain, the MMSE estimate of $\log X$ is $E[\log X∣Y=y]$. However, $\exp{E[\log X|Y=y]} \neq E[X|Y=y]$ rendering the log-transformation approach suboptimal. Therefore, these two denoising approaches are distinct. Our work correctly considers the bona fide multiplicative model without log transformation. We will include this discussion in the revised manuscript.
> We believe that these revisions satisfactorily clarify all the issues that you have raised: stronger theoretical foundation based on MLD with novel convergence analysis, distinction from prior art, additional experimental validation and suboptimality of the log transformation approach. In view of this rebuttal, we hope that this rebuttal is convincing and you will upgrade your score.
>
> Sketch:
> In the paper, we resort to the $\log$ transform and leverage the score change-of-variables formula to derive the reverse-time SDE. This choice was made to simplify the mathematics. Now, we demonstrate that we obtain the same reverse-time SDE starting from the original GBM SDE. Consider the GBM
> $$
> \mathrm{d}\boldsymbol{X}\_t = \mu \boldsymbol{X}\_t \,\mathrm{d}{t} + \sigma X\_t \,\mathrm{d}{\boldsymbol{W}\_t}
> $$
> and using the formula for the reverse-time SDE (Song et al. ICLR 2021), it can be shown that the reverse-time SDE is of the form
> $$
> \mathrm{d} \boldsymbol{X}\_t = \left((2\sigma^2\boldsymbol{1} - \boldsymbol{\mu}) X\_t + \sigma^2 \boldsymbol{X}\_t^2 \circ \nabla \log p_{\boldsymbol{X}\_t}(\boldsymbol{X}\_t, t)\right)\,\mathrm{d}{t} + \sigma X\_t \,\mathrm{d}{\boldsymbol{W}\_t}.
> $$
> At this juncture, we consider two possible discretizations. First, using Ito's lemma, we can show the reverse-time SDE can be transformed as
> $$
> \mathrm{d} \log \boldsymbol{X}\_t = \left(\left(\dfrac{3}{2}\sigma^2\boldsymbol{1} - \boldsymbol{\mu}\right) + \sigma^2 \boldsymbol{X}\_t \circ \nabla \log p_{\boldsymbol{X}\_t}(\boldsymbol{X}\_t, t)\right)\,\mathrm{d}{t} + \sigma \,\mathrm{d}{\boldsymbol{W}\_t},
> $$
> and this is equivalent to the reverse-time SDE mentioned in the main paper. The corresponding Euler-Maruyama discretization is given by
> $$
> \boldsymbol{X}\_{k-1} = \boldsymbol{X}\_k \circ \exp\left(- \delta \left(\boldsymbol{\mu} - \frac{3\sigma^2}{2}\boldsymbol{1}\right) + \delta\sigma^2 \boldsymbol{X}\_k \circ \nabla \log p_{\boldsymbol{X}\_k}(\boldsymbol{X}\_k, k) + \sqrt{\delta}\sigma \boldsymbol{Z}\_k\right).
> $$
> Alternatively, if we apply Euler-Maruyama discretization to the raw reverse-time SDE, we get
> $$
> \boldsymbol{X}\_{k-1} = \boldsymbol{X}\_k \left(\left(\boldsymbol{1} - \delta\boldsymbol{\mu} + 2\sigma^2\boldsymbol{1}\right) + \delta\sigma^2 \boldsymbol{X}\_k^2 \circ \nabla \log p_{\boldsymbol{X}\_k}(\boldsymbol{X}\_k, k) + \sqrt{\delta}\sigma \boldsymbol{Z}\_k\right).
> $$
> On comparing the two resulting discretizations, it is clear that both updates are multiplicative but only the one involving the exponential function preserves the sign of the entries of $\boldsymbol{X}\_k$ during sampling. The $\log$ transform through Ito's lemma preserves the sign during sampling and makes it easier to derive the reverse-time SDE. Through this exercise, we have also demonstrated that we don't really need the score change-of-variables formula (Robbins, 2024) to derive our result but leveraging it certainly simplifies the derivation and results in a sign-preserving update.
>
> # FID for Datasets
> |Methodology|MNIST|FMNIST|KMNIST|
> |----------------------------|-----|------|------|
> |DDPM (Ho et al NeurIPS 2020)|25.1526 | 48.0139| 54.7285|
> |DDIM ( Song et al ICLR 2021)| 28.1332 |13.5026|31.8864|
> |GBM(ours)|28.9616 |116.1499 |50.7832 |
>
>
>
>
> # KID for Datasets
> |Methodology|MNIST|FMNIST|KMNIST|
> |----------------------------|-----|------|------|
> |DDPM (Ho et al NeurIPS 2020)|0.0193 +/- 0.0007|0.0206+/-0.0011|0.0322+/-0.0016|
> |DDIM|0.0192+/-0.0019|0.1847+/- 0.0088|0.2671 +/- 0.0136|
> |GBM(ours)|0.02878 +/- 0.0015|0.4374 +/- 0.0044|0.0546 +/- 0.0021|
>
> # CIFAR-10 Results
>
> |Methodology|FID|
> |-----|-----|
> |DDPM (Ho et al NeurIPS 2020)|3.17|
> |DDIM ( Song et al ICLR 2021)|4.16|
> |GBM (ours) |98.73|
> |Poisson (Chen, T et al NeurIPS 2023) |4.80|
> |Beta (Zhou et al NeurIPS 2023) |3.66|

---

> > ### Comment · Reviewer_wtSy · 2025-08-02
> >
> > Thank you for your reply. The changes you describe help address many of my concerns, and I am leaning towards increasing my score to a 4.
> >
> > I want to clarify my question about theoretical analysis, as the result you report speaks to my question but does not fully address it. In the literature on sampling using Euclidean score matching, there are a variety of results that provide bounds on the quality of samples (e.g. in the sense of total variation distance) depending on the error in the score estimate (in an $L\_2$ sense, as in the regression loss used to train a denoising diffusion model), the discretization timestep, and the time horizon over which the forward process is run. For a representative result of this form, see Chen et al., "Sampling is as easy as learning the score: theory for diffusion models with minimal data assumptions", though there have been a host of more recent works giving refined bounds under varying assumptions. I am curious whether you think it would be challenging to prove an analogous result for the multiplicative models you study here. This question is also hinted at in the other referee's reports. I understand that proving bounds of this form may be beyond the scope of the present work, but it would be useful if you could at least comment upon the possibility.
> >
> > With regards to my question about discretization, I was mostly wondering about the numerical stability of the multiplicative updates. Concretely, if one wanted to prove a bound of the sort discussed above, would one in practice have to take floating-point precision into account? This is at least qualitatively related to the issues noted in work on numerical issues arising in the use of different models for hyperbolic space, see Mishne et al., "The Numerical Stability of Hyperbolic Representation Learning". The work on multiplicative low-precision updates you mention in your response to Reviewer KkH2 gives some hints in this direction, but does not quite get at the heart of what I'm asking.

---

> > > ### Author Response · Authors · 2025-08-04
> > > **Incorporating statements on theoretical bounds and numerical stability**
> > >
> > > **We’d like to thank you for your suggestions and for considering the possibility of updating your score. We hope that we’ve addressed your queries satisfactorily in the following response. We’d be happy to clarify any further concerns you may have.**
> > >
> > > We have gone through the paper by Chen et al. [1] that you’ve suggested as well as other related works  [2, 3, 4, 5], and we believe that it would be possible to derive similar bounds for multiplicative denoising score matching. This exercise does not seem to be straightforward and requires a thorough understanding of the results in [1–5] before developing similar results for the multiplicative setting. In fact, it would be insightful to see if we can possibly improve upon the bounds in certain settings (such as the non-negativity constraint) for certain classes of densities – log-concave, Poincaré and log-Sobolev inequalities. The MLD framework might make it easier to prove these theoretical bounds due to the generality of the framework. We will explore this interesting line of work.
> > >
> > > In alignment with your suggestion, we will add the following sentences in the Conclusions section of our paper.
> > >
> > > **``We proposed multiplicative denoising score matching as a means to train score networks for non-negative data corrupted by log-normal noise. Empirically, we showed that our model can generate high-quality samples from the target distribution. The empirical success of the proposed formalism can be further strengthened by providing appropriate theoretical guarantees along the lines of the analysis done by Chen et al. [1] and other researchers [2–5].’’**
> > >
> > > Thank you for bringing up the important issue of numerical precision and for sharing the work by Mishne et al. [6]. From what we’ve observed, most of the existing theoretical papers on sampling do not consider numerical precision effects in their analysis. Analysing the effect of numerical precision in the context of sampling algorithms is an interesting research direction in its own right and beyond the scope of our paper.
> > >
> > > Nishida et al. [7] propose multiplicative weight updates for training large neural networks, resulting in numerically stable low-precision updates. We work with sampling algorithms where our goal is to generate samples from the target distribution, whereas their work is concerned with optimisation of an objective over a dataset. We’ll have to study Nishida et al. [7] in greater detail to see if we can benefit from their analysis for our problem.
> > >
> > > In light of your suggestion, we will add the following sentences in section 4.4: Sampling details in the Supplementary part of our paper.
> > >
> > > **``Sampling with multiplicative updates using $\exp$ can lead to numerical overflow issues. We observed that larger values of step-size parameter $\delta$ caused overflow and the occurrence of NaN during sampling; and we overcame the instability issue by choosing a stable value for $\delta$ by performing a grid-search.’’**
> > >
> > >
> > >
> > > [1] Chen et al., Sampling is as easy as learning the score: theory for diffusion models with minimal data assumptions, ICLR 2023
> > >
> > > [2] Ahn et al., Efficient constrained sampling via the mirror-Langevin algorithm, NeurIPS 2023
> > >
> > > [3] Chewi et al., Analysis of Langevin Monte Carlo from Poincaré to log-Sobolev, COLT 2022
> > >
> > > [4] Balasubramanian et al., Towards a theory of non-log-concave sampling: first-order stationarity guarantees for Langevin Monte Carlo, COLT 2022
> > >
> > > [5] Zhang et al., Improved discretization analysis for underdamped Langevin Monte Carlo, COLT 2023
> > >
> > > [6] Mishne et al., The Numerical Stability of Hyperbolic Representation Learning, ICML 2023
> > >
> > > [7] Nishida et al., Log-Normal Multiplicative Dynamics for Stable Low-Precision Training of Large Networks, arxiv:2506.17768

---

> > > > ### Comment · Reviewer_wtSy · 2025-08-04
> > > >
> > > > Thank you for your response. Please be assured that I do of course understand that the setup of Nishida et al is distinct; I mentioned because it does to some degree speak to related issues of over/under-flow. I also understand that this is an issue on top of the sources of error considered in other sampling papers; my question is whether it is more severe with these multiplicative updates. I've also read through your discussion with Reviewer 8wzm and thought a bit more about the issue of technical novelty; I'm still not convinced by your arguments there. With this additional reflection, I'd encourage the authors to strengthen the biological angle of the manuscript and re-submit to a future conference once that and the additional experiments they mention in responses to other referees are completed. I think this paper could benefit from some further gestation.

---

> > > > > ### Author Response · Authors · 2025-08-05
> > > > > **Response to your assessment**
> > > > >
> > > > > Thank you.
> > > > > - The issues of overflow/underflow in sampling algorithms, in particular, our multiplicative sampling algorithm, is a novel research direction altogether and outside the scope of the current paper, as you rightly pointed out in your earlier response to our rebuttal. We do expect the issues to be more pronounced in the multiplicative setting but a complete characterization requires a separate line of investigation.
> > > > > - We had also indicated explicitly the new sentences that we would add to the manuscript in view of your suggestions.
> > > > > - Regarding our responses to Reviewer 8wzm, we'd like to understand concretely what your concerns there are, so we can attempt to clarify those for you.
> > > > >
> > > > > We believe that the paper has sufficient novelty.
> > > > >
> > > > > To the best of our knowledge, this is the first paper to propose Dale's Law inspired Generative models with multiplicative updates.
> > > > >
> > > > > Thanks to several suggestions by you (specifically, on Mirror Langevin Dynamics) and other reviewers, the paper has only become stronger on the theoretical front.
> > > > >
> > > > > Given that the discussion period has been extended by 48 hours, we would be happy to take any further questions.
> > > > >
> > > > > **We hope you will consider the contributions of the paper worthy of publication in NeurIPS 2025.**

---

> > > > > > ### Comment · Reviewer_wtSy · 2025-08-06
> > > > > >
> > > > > > Thanks for your additional response. For the reasons elaborated in my initial review, I do not find the motivation based on Dale's law compelling. I'd also remark that the MLD approach would correspond to running the sampling dynamics in the log domain (see eq. 3.2 and 3.3 of Hsieh et al); indeed, working in the dual space is critical for their analysis, which is based on porting over convergence results for unconstrained unadjusted Langevin dynamics. Absent a better-developed biological motivation, I still "computationally [...] do not see a reason why one would not use the log-transformed dynamics", to quote from my initial review.

---

> > > > > > > ### Author Response · Authors · 2025-08-07
> > > > > > > **On connections with Dale’s law, MLD and log-domain analysis**
> > > > > > >
> > > > > > > **Dale’s law connection** — The connection between Dale’s law and the proposed approach for diffusion is the “multiplicative update” rule for sampling. To the best of our knowledge, this is the first instance of a multiplicative noise based diffusion-driven generative model and a multiplicative update based sampler. This fascinating similarity is what we wish to motivate and highlight in our contribution. We do understand and acknowledge the differences — Dale’s law pertains to synaptic weights and the updates in our model correspond to pixel intensities. Historically, developments in artificial intelligence (AI) have been inspired by evidence of learning mechanisms in biological systems (perceptron, Boltzmann machines, CNNs, etc.) and the Dale’s law is one such important principle. The simplistic model of an artificial neuron that is all pervasive now in AI is no match to the sophisticated processing performed by a biological neuron. However, that did not stop the AI community from adopting it.
> > > > > > >
> > > > > > >
> > > > > > > **Link with MLD** — In our rebuttal, triggered by your question, we established a formal and non-obvious connection with MLD: the convex function in the Bregman divergence used to derive exponentiated gradient descent is identical to the one used to derive the Mirror Langevin Dynamics (MLD) SDE that leads to multiplicative updates for sampling. We also identified a choice of the convex function. We established a connection between the proposed reverse-time SDE and MLD SDE. This fascinating new discovery shows that our bio-inspired approach is a principled approach to discovering a new class of generative models, opening new avenues for future work.
> > > > > > >
> > > > > > >
> > > > > > > **Operating in the Log Domain** — We have proved conclusively in the rebuttal and discussion phase that the proposed approach is NOT log-domain DDPM. The SDEs for log-domain DDPM and GBM are different and consequently, their discretised versions also lead to different updates. Please see our response to Reviewer 8wzm. In fact, Reviewer 8wzm has rightly pointed out that exponential gradient descent is NOT equivalent to gradient descent in the log domain. Denoising is at the heart of diffusion-based generative modeling. We showed that log-domain denoising is suboptimal to operating directly in the multiplicative space. Please see our rebuttal to Reviewer 8wzm under the heading “It’s Not just Diffusion in the Log Domain” and response to Reviewer 8wzm “GBM is not log-domain DDPM.” Reviewer 8wzm also acknowledges the technical differences between the two frameworks.
> > > > > > >
> > > > > > > **We hope you find these responses convincing. We’d be happy to answer any further questions you may have.**

---

> > > > > > > > ### Author Response · Authors · 2025-08-09
> > > > > > > >
> > > > > > > > Dear Reviewer,
> > > > > > > >
> > > > > > > > Thank you for your insightful feedback, suggestions, and active engagement in the discussion phase. We hope you find the paper acceptable for publication. If you have any further questions, we will be happy to offer clarifications.
> > > > > > > >
> > > > > > > > Thank you.

---

### Official Review · Reviewer_8wzm · 2025-07-02

**Clarity:** 4
**Significance:** 1
**Originality:** 3
**Rating:** 3
**Confidence:** 4

**Summary:**

This paper introduces a variant of diffusion models driven by geometric Brownian motion (with multiplicative rather than additive updates, the forward process converging to a log-normal distribution). The resulting equations are similar to exponential gradient descent in optimization, which has been proposed as a more biologically plausible learning algorithm that respects Dale's law (that signs of parameters cannot change during training). The proposed method is evaluated numerically on the MNIST, Fashion-MNIST, and Kuzushiji-MNIST datasets.

**Questions:**

- Is the proposed approach equivalent to regular diffusion in log space?
- Can the authors detail the potential biological relevance of the proposed algorithm?

I am willing to change my score if the authors convincingly explain how their approach is significantly novel or impactful to stand out among the many diffusion variants in the literature.

**Ethical Concerns:**

["NO or VERY MINOR ethics concerns only"]

**Final Justification:**

The authors did not demonstrate significant differences between their approach and standard diffusion, nor a significant biological relevance.

**Limitations:**

As admitted by the authors, the quality of the generated samples is subpar, and the experiment validation is light.

**Quality:**

2

**Strengths And Weaknesses:**

The paper is very clearly written, the proposed idea is well motivated and cleanly implemented. My main criticism of the paper is that its framework can be summarized in one sentence: "diffusion models in log domain". As noted in the paper, the forward and backward SDEs are the usual ones in the log domain (eqs. (9) and (10)), and as mentioned by Hyvärinen in [25], the data-weighted score matching objective (eq. (21)) is in fact equivalent to regular score matching in log space. The only difference that I can see is that discretization of the SDEs in image space and log space are not equivalent (the former leads to additive updates and the latter multiplicative updates when expressed in image space), but it is clear that one should use the latter if one wants to ensure positivity.

There are already many similar variants of diffusion models in the literature, see e.g.:
- with Gamma noise (ensures positivity): Nachmani, E., Roman, R. S., & Wolf, L. (2021). Denoising diffusion gamma models. arXiv preprint arXiv:2110.05948.
- with Beta noise (pixel values remain in [0, 1]): Zhou, M., Chen, T., Wang, Z., & Zheng, H. (2023). Beta diffusion. Advances in Neural Information Processing Systems, 36, 30070-30095.
- with Poisson noise (ensures positivity, discrete domain): Chen, T., & Zhou, M. (2023, July). Learning to jump: Thinning and thickening latent counts for generative modeling. In International Conference on Machine Learning (pp. 5367-5382). PMLR.
- with general constraints (e.g. data positivity): Fishman, N., Klarner, L., De Bortoli, V., Mathieu, E., & Hutchinson, M. (2023). Diffusion models for constrained domains. arXiv preprint arXiv:2304.05364.

and many others ("soft diffusion", "cold diffusion", "Riemannian diffusion", "discrete diffusion", etc).

While the geometric/log domain diffusion proposed here is novel, to the best of my knowledge, the question is what makes this specific variant stand out from all the others. One possible answer could be that it leads to better generative models with improved performance (be it image quality, number of sampling steps, training efficiency, etc), but it does not seem to be the authors' goal. In fact, the experimental results are arguably quite weak: the three datasets are small (60k images at 28x28 resolution) and image quality is poor (MNIST samples have visible artifacts, Fashion-MNIST samples lack contrast, and Kuzushiji-MNIST samples do not look very realistic to my admittedly uneducated eye). While I am in favor of dropping SOTA at largest scale requirements for novel methods, there needs to be another reason to justify its potential impact. The authors seem to be motivated by biological plausibility in connection to Dale's law. This is not my area of expertise, but I fail to see the connection here. I understand the motivation for exponential gradient descent as a biologically plausible learning algorithm, but it is not clear to me why the brain would implement diffusion on sign-locked synaptic weights.

As a side note, EGD is not equivalent to GD in the log domain. To obtain structurally similar equations for EDG and geometric diffusion, the authors have to introduce an additional change of variable. This raises the question whether the proposed geometric diffusion is the natural counterpart to EGD, or if there is another formulation.

---

> ### Author Rebuttal · Authors · 2025-07-31
>
> Thank you for the thoughtful and detailed review. We're delighted you found our paper "very clearly written" and the idea "well motivated," and we appreciate your explicit willingness to reconsider your score.
> Your main concern seems to be that our framework is an oversimplification of "diffusion models in log domain." We appreciate the chance to clarify this crucial point. We believe this view obscures the key theoretical and practical contributions of our work, which we'll detail below alongside new experimental results.
> - __It's Not Just Diffusion in the Log Domain__
>     - __Our approach is fundamentally different from standard diffusion in the log domain, both mathematically and practically__.
> As you've rightly pointed out, "EGD is not equivalent to GD in the log domain." This same core principle applies to the proposed Geometric Brownian Motion (GBM) framework. It is indeed not the same as applying a standard DDPM in log space. The distinction is subtle, but critical.
>     - __The update rules are different__. A standard DDPM applied to log-transformed data results in a reverse-time SDE that is structurally and functionally different from the one derived in the paper. A detailed mathematical derivation to this effect is included below.
>     - __Our formulation is a principled design choice__. As you correctly note, the choice of discretization is crucial. Our method yields principled multiplicative updates and the non-negativity constraint is satisfied naturally.
>     - __MMSE Denoiser property__. Denoising is at the heart of diffusion-based generative modelling. To appreciate the distinction further, consider Minimum Mean Squared Error (MMSE) estimators for the multiplicative noise setting with and without log transformation. Given the forward model $Y=XN$, $X$ = signal, $N$=noise, and the noisy measurement $Y=y$, the optimal/MMSE estimate of $X$ is the conditional expectation $E[X∣Y=y]$. If instead one works in the log domain, the MMSE estimate of $\log X$ is $E[\log X∣Y=y]$. However, $\exp{\left(E[\log X|Y=y]\right)} \neq E[X|Y=y]$ which can also be corroborated by Jensen's inequality. Therefore, these two denoising approaches are distinct. Our work correctly considers the true multiplicative model.
> - **Novelty and Impact**
> Our work's novelty and impact lie in providing the first SDE-based formulation for multiplicative score matching and establishing a new, deep connection between biologically plausible optimization and generative modeling.
>     - **A Principled SDE Framework for Non-Negative Data**: We're the first to derive from first principles a novel score-matching formulation which also explains Hyvärinen’s heuristic and computationally intractable, non-negative explicit score matching loss.
>     - **A New Tractable Loss**: We leverage this link to derive the Multiplicative Denoising Score Matching (M-DSM) loss and formally prove its equivalence to Hyvärinen’s intractable explicit score-matching loss.
>     - **A New Bridge Between Sampling and Optimization**: Our work is the first to establish a formal link between a generative sampling process (the GBM SDE) and a biologically-inspired optimization algorithm, namely, Exponentiated Gradient Descent. This connection is a mathematical identity via the lens of Mirror Langevin Dynamics (MLD).
>     - **New Convergence Guarantees**: We provide two convergence proofs: one showing that the forward and reverse Fokker-Planck Equations match (guaranteeing correct sampling) and another, via MLD, showing convergence for the log-normal case. Please refer to our response to reviewer VjUr (we don't include this here due to space constraints).
> - **On Biological Relevance**
> The biological motivation is an analogy for principled constraint satisfaction, not a literal model of brain function.
> We do not posit that the brain implements diffusion on synaptic weights. Rather, Dale's Law encapsulates biological evidence of learning without synaptic flips. This inspired us to build a generative sampler with an analogous sign-preserving property (i.e., non-negativity for images) using multiplicative updates.
> This analogy led us to a framework whose validity is confirmed by a formal mathematical link. The convex function in the Bregman divergence that yields Exponentiated Gradient Descent is identical to the one that defines the Mirror Langevin Dynamics sampler that matches our SDE. This shows our biologically-inspired principle leads directly to a powerful and theoretically sound generative model.
> - **New Experimental Results**
> Following your suggestions, we've conducted new experiments showing the performance of our model on standard baselines and competitive with other relevant constrained-domain diffusion models.
> To address your concerns about our experimental results, we performed several new experiments for performance benchmarking:
> In the following Table, we compare our method against DDPM and DDIM on MNIST, Fashion-MNIST, and Kuzushiji-MNIST. Although the FIDs are not clearly outperforming the state-of-the-art the visual quality of the generated images is good indicating that the proposed technique is promising and can be further fine-tuned to improve FID.
>
> We extended our evaluation to CIFAR-10 and obtained an FID of 98.73. Our performance is below par compared to the models you mentioned, including Beta Diffusion and Poisson Diffusion. Within the time constraints of the rebuttal, our models could not be fully optimized to produce the best FID. However, the visual quality of generated samples is high and indicates that there is significant room for improving upon the FID values. We will include the images and the final FID scores in the supplementary of the manuscript.
> We hope these clarifications, new theoretical links, and new experimental results convincingly demonstrate that our contribution is novel and impactful. We would be grateful if you could consider these aspects and upgrade your score.
>
> # FID for Datasets
> |Methodology|MNIST|FMNIST|KMNIST|
> |----------------------------|-----|------|------|
> |DDPM (Ho et al NeurIPS 2020)|25.1526 | 48.0139| 54.7285|
> |DDIM ( Song et al ICLR 2021)| 28.1332 |13.5026|31.8864|
> |GBM(ours)|28.9616 |116.1499 |50.7832 |
>
>
>
>
> # KID for Datasets
> |Methodology|MNIST|FMNIST|KMNIST|
> |----------------------------|-----|------|------|
> |DDPM (Ho et al NeurIPS 2020)|0.0193 +/- 0.0007|0.0206+/-0.0011|0.0322+/-0.0016|
> |DDIM|0.0192+/-0.0019|0.1847+/- 0.0088|0.2671 +/- 0.0136|
> |GBM(ours)|0.02878 +/- 0.0015|0.4374 +/- 0.0044|0.0546 +/- 0.0021|
>
> # CIFAR-10 Results
>
> |Methodology|FID|
> |-----|-----|
> |DDPM (Ho et al NeurIPS 2020)|3.17|
> |DDIM ( Song et al ICLR 2021)|4.16|
> |GBM (ours) |98.73|
> |Poisson (Chen, T et al NeurIPS 2023) |4.80|
> |Beta (Zhou et al NeurIPS 2023) |3.66|

---

> > ### Comment · Reviewer_8wzm · 2025-08-04
> >
> > I am not convinced by the differences between GBM and diffusion in the log domain. Your reply mentions "a detailed derivation" that I couldn't find. Besides, I understand that E[X | Y] and E[log X | Y] are different, but this is just a statement that regular diffusion (which adds Gaussian noise to X, so that E[X | Y] is connected to the score of X) is different from GBM (which adds Gaussian noise to log X, so that E[log X | Y] is connected to the score of log X--which is itself related to the score of X via a simple changes of variables). Don't equations (8) and (10) directly show that the forward and backward SDEs are the usual standard linear-Gaussian SDEs?

---

> ### Author Response · Authors · 2025-08-04
> **GBM is not log-domain DDPM**
>
> Thank you for your assessment. Due to space constraints, we included the derivation in response to reviewer KkH2. We reproduce it here. Starting from non-negative data, applying log and using DDPM leads to a reverse-time SDE that is distinct from the proposed GBM approach.
> While the score of $X$ and the score of $\log(X)$ are related through a change of variables formula, we observed experimentally that plugging in a pre-trained score network did not result in generation -- this warrants the score matching formalism proposed in our paper.
>
> 1. Consider the GBM SDE
> $$ \mathrm{d}\boldsymbol{X}\_t = \mu \boldsymbol{X}\_t \mathrm{d}{t} + \sigma \boldsymbol{X}\_t \mathrm{d}{\boldsymbol{W}\_t}$$
> and using Anderson's result [2], the reverse-time SDE takes the form
> $$\mathrm{d} \boldsymbol{X}\_t = \left((2\sigma^2\boldsymbol{1} - \boldsymbol{\mu}) \circ \boldsymbol{X}\_t + \sigma^2 \boldsymbol{X}\_t^2 \circ \nabla \log p\_{\boldsymbol{X}\_t}(\boldsymbol{X}\_t, t)\right)\mathrm{d}{t} + \sigma \boldsymbol{X}\_t \mathrm{d}{\boldsymbol{W}\_t}.$$
> At this juncture, there are at least two possible approaches. First, using Ito's lemma, we can show the reverse-time SDE can be transformed as
> \begin{equation}
>     \mathrm{d} \log \boldsymbol{X}\_t = \left(\left(\dfrac{3}{2}\sigma^2\boldsymbol{1} - \boldsymbol{\mu}\right) + \sigma^2 \boldsymbol{X}\_t \circ \nabla \log p\_{\boldsymbol{X}\_t}(\boldsymbol{X}\_t, t)\right)\mathrm{d}{t} + \sigma \mathrm{d}{\boldsymbol{W}\_t},
> \end{equation}
> which matches the reverse-time SDE given in the paper. The corresponding Euler-Maruyama discretization is given by
> $$\boldsymbol{X}\_{k-1} = \boldsymbol{X}\_k \circ \exp\left(- \delta \left(\boldsymbol{\mu} - \frac{3\sigma^2}{2}\boldsymbol{1}\right) + \delta\sigma^2 \boldsymbol{X}\_k \circ \nabla \log p\_{\boldsymbol{X}\_k}(\boldsymbol{X}\_k, k) + \sqrt{\delta}\sigma \boldsymbol{Z}\_k\right),$$ which is sign-preserving.
>
> Alternatively, if we apply Euler-Maruyama discretization to the SDE without the log transform, we get
> $$\boldsymbol{X}\_{k-1} = \boldsymbol{X}\_k \left(\left(\boldsymbol{1} - \delta\boldsymbol{\mu} + 2\sigma^2\boldsymbol{1}\right) + \delta\sigma^2 \boldsymbol{X}\_k^2 \circ \nabla \log p\_{\boldsymbol{X}\_k}(\boldsymbol{X}\_k, k) + \sqrt{\delta}\sigma \boldsymbol{Z}\_k\right).$$
> On comparing the two discretizations, it is clear that both updates are multiplicative but only the former preserves the sign of the entries of $\boldsymbol{X}\_k$ during sampling. This exercise demonstrates that one doesn't really need the score change-of-variables formula to derive our result but leveraging it certainly simplifies the derivation and results in a sign-preserving update.
>
> 2. We show here that DDPM in the log domain results in a different reverse-time SDE than the one proposed in our paper.
>
> The standard DDPM forward SDE (Eq. 11 from Song et al. [1]) is:
> $$\mathrm{d}\mathbf{Y}\_t = -\dfrac{\beta(t)}{2}\mathbf{Y}\_t \mathrm{d}{t} + \sqrt{\beta(t)} \mathrm{d}{\mathbf{W}\_t}$$where $\beta(t)$ is the variance schedule. The corresponding reverse-time SDE is:$$\mathrm{d}\mathbf{Y}\_t = \left(\dfrac{\beta(t)}{2}\mathbf{Y}\_t + \beta(t)\nabla\_{\mathbf{Y}\_t}\log p(\mathbf{Y}\_t)\right)\mathrm{d}{t} + \sqrt{\beta(t)} \mathrm{d}{\mathbf{W}\_t}$$
> Now, substituting $\mathbf{Y}\_t = \log \mathbf{X}\_t$ and using the score change-of-variables formula to express the score $\nabla\_{\mathbf{Y}\_t}\log p(\mathbf{Y}\_t)$ in terms of the score of the original data, $\nabla\_{\mathbf{X}\_t}\log p(\mathbf{X}\_t)$, we get the reverse-time SDE in terms of $\mathbf{X}\_t$:$$
> \mathrm{d}\log \mathbf{X}\_t = \left(\dfrac{\beta(t)}{2}\log\mathbf{X}\_t + \beta(t)\left(\mathbf{1} + \mathbf{X}\_t \circ \nabla\_{\mathbf{X}\_t} \log p(\mathbf{X}\_t)\right)\right)\mathrm{d}{t} + \sqrt{\beta(t)} \mathrm{d}{\mathbf{W}\_t}
> $$On Euler-Maruyama discretization with step size $\delta$ and schedule $\beta\_k = \beta(k\delta)$, the update rule becomes:$$
> \mathbf{X}\_{k-1} = \mathbf{X}\_k^{\left(1 + \frac{\delta\beta\_k}{2}\right)} \circ \exp\left( \delta\beta\_k\left(\mathbf{1} + \mathbf{X}\_k \circ \nabla\_{\mathbf{X}\_k} \log p(\mathbf{X}\_k)\right) + \sqrt{\delta\beta\_k} \mathbf{Z}\_k\right)$$ This is also a sign-preserving update, but, this is structurally different from the sampler derived in our paper due to the additional exponent factor of $\left(1+  \frac{\delta\beta_k}{2}​​\right)$ on $\mathbf{X}\_k$.
>
> __The preceding analysis confirms that our framework is not simply log-domain DDPM.__
> **A term-by-term comparison of Eq.(8) in our paper with Eq.(11) in Song et al. [1] also shows that they are NOT identical.**
> Thank you for the question. We will include this analysis in the paper.
>
> __In view of this analysis, we hope you will consider increasing your score.__
>
> [1] Song et al., Score-Based Generative Modeling Through Stochastic Differential Equations, ICLR 2021
>
> [2] Anderson, Reverse-time diffusion equation models, Stochastic Processes and their Applications 1982

---

> > ### Comment · Reviewer_8wzm · 2025-08-04
> >
> > Thank you for your reply. It is clear that the only "differences" with diffusion in the log domain are indeed the discretization and a rather simple change of linear drift. I don't consider these changes to be significant enough for GBM to stand as a "fundamentally different" alternative to standard diffusion. Given the weakness of the experimental results and the biological motivation, I maintain my score.

---

> > > ### Author Response · Authors · 2025-08-05
> > > **Geometric Brownian Motion is fundamentally different from standard diffusion!**
> > >
> > > **We are glad you are finally convinced that the proposed approach is not the same as log-domain DDPM.**
> > >
> > > All stochastic differential equations (SDEs) are characterised by a drift and a diffusion component. What distinguishes one SDE from another is the choice of the drift and diffusion terms. These distinctions also show up in the corresponding discretizations leading to different sampling algorithms as we have demonstrated.
> > >
> > > GBM is **fundamentally different** from arithmetic Brownian motion, which is what led to its widespread adoption in areas such as mathematical finance. To the best of our knowledge, ours is the first paper to propose GBM-based Generative models with multiplicative updates for sampling.
> > >
> > > Thanks to your comments and other reviewers, we also have several results that strengthen our paper as mentioned in our responses during the discussion phase.
> > >
> > > Given that the discussion period has been extended by 48 hours, we would be happy to take any further questions.

---

> > > > ### Author Response · Authors · 2025-08-09
> > > >
> > > > Dear Reviewer,
> > > >
> > > > Thank you for your insightful feedback, suggestions, and active engagement in the discussion phase. We hope you find the paper acceptable for publication. If you have any further questions, we will be happy to offer clarifications.
> > > >
> > > > Thank you.

---

### Official Review · Reviewer_VjUr · 2025-07-04

**Clarity:** 2
**Significance:** 1
**Originality:** 3
**Rating:** 4
**Confidence:** 3

**Summary:**

The paper proposes a method that introduces a biologically inspired twist (with roots also in Geometric Brownian Motion) to score-based generative models. The authors build on exponential gradient descent (EGD) and show that the discretisation of the reverse-time SDE yields a multiplicative resampling rule structurally equivalent to EGD.  The method is then evaluated on MNIST, Fashion-MNIST, and Kuzushiji MNIST, but not compared with other denoising diffusion state-of-the-art models (DDPM, DDIM).

**Questions:**

- Why is your method preferable to using a specialized noise scheduling scheme?
- Why is it difficult to derive an error bound for your proposed paradigm?
- In the limitations section (line 318), you mention that the method requires a large amount
  of training data and computational resources. Can you quantify this?
- What is the actual computational cost of your method?
- Compared to classical DDPM methods, what is the additional computational overhead introduced
  by your approach?

**Ethical Concerns:**

["NO or VERY MINOR ethics concerns only"]

**Final Justification:**

I think the work is interesting and could trigger a lot of fruitful discussion at NeurIPS. However, the Empirics still feel a bit weak, unfortunately, especially in comparison to the rest of my reviewing batch, so I have to rate somewhat relatively across the works/standards I am seeing.

Edit: I have updated the score again, the conceptual aspect of this work is novel and promising even though the empirics lack the authors are pushing towards stronger ablations/explorations for the final camera ready.

**Limitations:**

While the paper is interesting (because of the connection the proposed method has with neuroscience):
- it does not theoretically motivate why the method is superior to others
- it does not show empirically that the method is superior to others
- and it requires a large amount of data and computational resources, with no mention of how large these should be

I would be encouraged to raise the score if:
- some form of benchmarking were included
- there were a theoretical foundation explaining why the method should perform better

An application to financial prediction might also help illustrate the practical value of the approach  .

**Quality:**

2

**Strengths And Weaknesses:**

Major Strengths:
- The main result of the paper, Theorem 5.1, highlights a surprising and thought-provoking connection between Dale’s Law and Geometric Brownian Motion. I found this link both original and interesting. Overall, the paper is well-written and well-structured. The authors do a nice job of illustrating the connection between neuroscience and stochastic calculus in a way that feels intuitive and engaging.

Major Weaknesses:
- Empirical: The proposed method is not compared against other denoising diffusion models such as DDPM or DDIM. Including such comparisons would help clarify the practical benefits of the approach, especially in settings where computational resources are not a limiting factor. For Figure 2 (Fashion MNIST), there appear to be signs of mode collapse, though it’s difficult to say definitively without side-by-side comparisons. Similarly, for Kuzushiji MNIST, it would be helpful to include reference samples or evaluation metrics to assess whether the generated characters are meaningful. Adding these elements would perhaps strengthen the empirical section of the paper.
- Theoretical: Given that the empirical results are somewhat limited, it would be helpful to include a theoretical justification or guarantee that the proposed method leads to improved performance, e.g., reduced integration error or better score approximation. Without such analysis, it’s difficult to assess whether the approach is expected to consistently outperform existing methods.
- Besides the mention of the fact that the method requires large training data and computation resources, no further exploration is done.
- Financial application (prediction of markets is presented as a field where the method could be applied), but no experiments in that regard (conclusion).
Minor Weaknesses:
- No error bars for FID/KID
- Authors mention that the hyperparameter search was “carefully” done (322). What do they mean?

---

> ### Author Rebuttal · Authors · 2025-07-31
>
> We are grateful to Reviewer VjUr for the constructive and actionable feedback. We were particularly encouraged that you found the core connection between Dale's Law and Geometric Brownian Motion to be surprising, original, and interesting.
> We have followed your suggestions and believe that revising the manuscript accordingly will make it substantially stronger. We present new benchmarking results against DDPM/DDIM, further clarify the theoretical foundations that motivate our method's advantages, and provide concrete responses to all other points raised.
> 1. **On Empirical Evaluation & Benchmarking**
> We have added new benchmarking experiments against DDPM and DDIM
> New Benchmarks and Scaling to Color Images: We have now conducted these experiments and report the results in the below. We also extended our evaluation to CIFAR-10 and obtained an FID of 98.73. Our performance is below par compared to the models you mentioned, including Beta Diffusion and Poisson Diffusion. Within the time constraints of the rebuttal, our models could not be fully optimized to produce the best FID. However, the visual quality of generated samples is high and indicates that there is significant room for improving upon the FID values. We will include the images and the final FID scores in the supplementary of the manuscript.
> Mode Collapse & Sample Quality Figure 2:  This is a fair observation from visual inspection and we are investigating the cause of the mode collapse during sampling. Further optimization of the step-size, choice of noise scheduler could potentially avoid this issue.
> Error Bars: We followed the standard practice of not reporting the error bars on FID like many leading papers on diffusion modelling do(e.g., DDPM, DDIM). Since FID is reported on a large set of 50k generated images, the variance is typically very low.
> 2. **On Theoretical Justification & Error Bounds**
> Our method's advantage stems from providing the first SDE-based formulation for multiplicative score matching, establishing a new, deep connection between bio-inspired optimization and generative modeling and a newfound connection to Mirror Langevin Dynamics, which offers a path toward formal guarantees.
> Why It Should Perform Better: Our framework is not an arbitrary choice but is specifically tailored for non-negative data like images. Standard DDPMs use additive Gaussian noise, which can push pixel values below zero, forcing the model to learn a difficult "clamping" behavior at the boundary. Our multiplicative GBM process naturally respects the non-negativity constraint at every step of the SDE, making the learning task for the score network simpler and more direct.
> Connection to Mirror Langevin Dynamics (MLD): The theoretical "why" is further strengthened by a new insight we gained during the rebuttal period. Our reverse-time SDE is related to Mirror Langevin Dynamics, a powerful class of samplers designed for constrained domains. The convex function used in the Bregman divergence that underpins  EGD updates is precisely the one used to derive this specific MLD sampler. This is yet another theoretical link, connecting our biologically-inspired method to a well-studied family of optimal transport and sampling algorithms.
> On Error Bounds: Leveraging the MLD link, we have derived convergence behaviour of generated samples for a lognormal target distribution that evolves with the MLD SDE. The MLD connection allows us to provide the kind of theoretical analysis you were looking for. We now include a proof that for a log-normal target distribution $\mathcal{LN}(\mathbf{\mu}, \sigma^2 \mathbf{I})$, the discretized dynamics of the MLD equation above converge to a limiting measure $\mathcal{LN}\left(\mathbf{\mu}, \frac{\sigma^2}{1 - \delta/(2\sigma^2)} \mathbf{I}\right)$. This analysis not only proves convergence to the correct mean but also precisely quantifies the discretization error in the covariance, which is an altogether new theoretical contribution of the paper significantly elevating the technical content.
> 3. **On Computational Cost & Hyperparameters**
> Our method has the same computational complexity as a standard DDPM of equivalent size, and we now provide precise training details and clarify our hyperparameter search.
> Computational Overhead: There is no additional computational overhead compared to DDPM. The primary cost during sampling is the number of function evaluations (NFE) of the score network. Since both our method and DDPM use the same number of reverse-time steps (N=1000), the sampling cost for a network of the same size is identical.
> Quantified Resources: We have already mentioned these details in section 4.3 of the supplementary document.
> Hyperparameter Search: For sampling, the quality depends on the reverse-time step size. We performed a grid search over step sizes in the range [10−8,10−3], selecting the configuration that produced the best sample quality, which we now quantify with FID and KID.
> 4. **Other Clarifications**
> On Specialized Noise Schedules: The proposed approach is not dependent on the choice of noise schedule and just like DDPM, our framework can be combined with any noise scheduling scheme (e.g., linear, cosine) to potentially improve results further. However, a thorough investigation pertaining to the optimal noise schedules in the multiplicative setting remains an open issue.
> On Financial Applications: Incidentally, during the rebuttal phase we came across a preprint by Kim et al. “A diffusion-based generative model for financial time series via geometric Brownian motion, https://arxiv.org/pdf/2507.19003, uploaded on 25th July, 2025.” They propose a diffusion-based generative framework for financial time series that incorporates GBM into the forward noising process, however, unlike our approach, they effectively deal with an additive noise model by using the log transform. There seems to be compelling evidence building up to show the efficacy of log normal noise and GBM SDE for generative modeling.
> We are confident that these new experiments, deeper theoretical framing, and detailed clarifications directly address the points you raised. We have provided the requested benchmarking and a theoretical foundation for our method's advantages. We hope you'll agree that these additions substantially strengthen the paper and provide the justification needed to reconsider your score.
>
> # FID for Datasets
> |Methodology|MNIST|FMNIST|KMNIST|
> |----------------------------|-----|------|------|
> |DDPM (Ho et al NeurIPS 2020)|25.1526 | 48.0139| 54.7285|
> |DDIM ( Song et al ICLR 2021)| 28.1332 |13.5026|31.8864|
> |GBM(ours)|28.9616 |116.1499 |50.7832 |
>
>
>
>
> # KID for Datasets
> |Methodology|MNIST|FMNIST|KMNIST|
> |----------------------------|-----|------|------|
> |DDPM (Ho et al NeurIPS 2020)|0.0193 +/- 0.0007|0.0206+/-0.0011|0.0322+/-0.0016|
> |DDIM|0.0192+/-0.0019|0.1847+/- 0.0088|0.2671 +/- 0.0136|
> |GBM(ours)|0.02878 +/- 0.0015|0.4374 +/- 0.0044|0.0546 +/- 0.0021|
>
> # CIFAR-10 Results
>
> |Methodology|FID|
> |-----|-----|
> |DDPM (Ho et al NeurIPS 2020)|3.17|
> |DDIM ( Song et al ICLR 2021)|4.16|
> |GBM (ours) |98.73|
> |Poisson (Chen, T et al NeurIPS 2023) |4.80|
> |Beta (Zhou et al NeurIPS 2023) |3.66|
>
> Convergence through the Fokker-Planck equations.
> For the forward SDE
> $$
> \mathrm{d}\boldsymbol{Y}\_t = \left(\boldsymbol{\mu} - \frac{\sigma^2}{2}\boldsymbol{1}\right) \,\mathrm{d}{t} + \sigma \,\mathrm{d}{\boldsymbol{W}\_t},
> $$
> the corresponding Forward FPE is
> $$
> \dfrac{\partial p_{\boldsymbol{Y}\_t}(\boldsymbol{X}\_t, t)}{\partial t} = - \nabla \cdot \left(\left(\boldsymbol{\mu} - \frac{\sigma^2}{2}\boldsymbol{1}\right) p_{\boldsymbol{Y}\_t}(\boldsymbol{Y}\_t, t)\right) + \frac{\sigma^2}{2}\Delta^2 p_{\boldsymbol{Y}\_t}(\boldsymbol{Y}\_t, t). \tag{1}
> $$
> The reverse-time SDE is
> $$\mathrm{d}\boldsymbol{Y}\_t = -\left(\boldsymbol{\mu} - \frac{\sigma^2}{2}\boldsymbol{1} - \sigma^2 \nabla \log p_{\boldsymbol{Y}\_t}(\boldsymbol{Y}\_t, t)\right) \,\mathrm{d}{t} + \sigma \,\mathrm{d}{\boldsymbol{W}\_t},$$
> and its corresponding reverse-time FPE is given by
> $$ -\dfrac{\partial p_{\boldsymbol{Y}\_t}(\boldsymbol{X}\_t, t)}{\partial t} = \nabla \cdot \left(\left(\boldsymbol{\mu} - \frac{\sigma^2}{2}\boldsymbol{1} - \sigma^2 \nabla \log p_{\boldsymbol{Y}\_t}(\boldsymbol{Y}\_t, t)\right) p_{\boldsymbol{Y}\_t}(\boldsymbol{Y}\_t, t)\right) + \Delta^2\left(\frac{\sigma^2}{2} p_{\boldsymbol{Y}\_t}(\boldsymbol{Y}\_t, t)\right).$$
> Since $\nabla \cdot \nabla \log p_{\boldsymbol{Y}\_t}(\boldsymbol{Y}\_t, t) = \Delta^2 p_{\boldsymbol{Y}\_t}(\boldsymbol{Y}\_t, t)$~\citep{Lyu_score_09}, we can rewrite the reverse-time FPE as
> $$
> \dfrac{\partial p_{\boldsymbol{Y}\_t}(\boldsymbol{X}\_t, t)}{\partial t} = -\nabla \cdot \left(\left(\boldsymbol{\mu} - \frac{\sigma^2}{2}\boldsymbol{1}\right)p_{\boldsymbol{Y}\_t}(\boldsymbol{Y}\_t)\right) + \frac{\sigma^2}{2}\Delta^2 p_{\boldsymbol{Y}\_t}(\boldsymbol{Y}\_t, t). \tag{2}
> $$
> Comparing Eq. (1) and Eq. (2), we see that they are identical. The reverse-time SDE is designed to generate samples from the target distribution and we refer the reader to (Anderson1982) for a detailed proof.

---

> > ### Comment · Reviewer_VjUr · 2025-08-03
> > **Quick followup**
> >
> > I have raised my score, my understanding of the paper has improved and it is indeed quite interesting, at the conceptual level I do feel this work has a lot of value, however some remaining concerns / questions
> >
> > 1. Looking at the provided FIDs it does seem the approach is worse than DDPM most of the time ?
> > 2. At the end of your rebuttal you are providing a semi-heruistic standard argument , justifying the standard timer-reversal result and pointing me to Andersons seminal work, I am familiar / ok with these results (I don't have a problem with them), what point of my review are they supposed to be addressing ?

---

> > ### Author Response · Authors · 2025-08-04
> > **Clarifications on FID and theoretical contributions**
> >
> > Thank you for re-evaluating our paper and its contributions and increasing your score in light of the rebuttal.
> >
> > 1. Yes, although visually the generated samples seem to be of similar quality as the training data and FID obtained by GBM model is better than DDPM and DDIM for Kuzushiji MNIST, the GBM FIDs are slightly worse than DDPM and DDIM for MNIST and much worse than DDPM and DDIM for Fashion MNIST. To further improve the FIDs, we are also experimenting with a final denoising step at the end of sampling as proposed by Song et al. [1].
> > Within the time constraints of the rebuttal, we could not further optimise the model performance in terms of FID. We believe that there is potential to improve the FID by optimising the architecture, fine-tuning the hyperparameters, applying exponential moving average (EMA), etc., during sampling.
> > 2. This is an additional finding on the theoretical front in support of our work that we found interesting and wanted to bring to your attention. This result will enable us to derive GBM-based generative models in other frameworks, such as flow-matching and variational auto-encoders with the non-negativity constraint.
> >
> > Further, substantiating our response to your question on the theoretical contribution:
> > The score matching formulation used in the additive noise setting [1, 2] is neither appropriate nor applicable to the multiplicative case considered in our paper. Our SDE-based formulation allows us to derive a principled score-matching formulation together with a practically viable strategy. Our formulation also presents a mathematical foundation on which Hyvaerinen’s heuristically motivated result [3] can be interpreted for score matching of non-negative data. While Hyvaerinen’s concept is based on a continuity argument, our derivation is based on a principled SDE approach. This is one of our major theoretical contributions.
> >
> > We posit that similar extensions can be developed for other multiplicative noise models, for instance, the multiplicative Gamma distributed noise encountered in real-world imaging situations such as synthetic aperture RADAR and optical coherence tomography.
> >
> > We would be happy to address any further questions you may have.
> >
> > [1] Song et al., Improved Techniques for Training Score-Based Generative Models, NeurIPS 2020
> > [2] Hyvaerinen, Estimation of Non-Normalized Statistical Models by Score Matching, JMLR 2005
> > [3] Hyvaerinen, Some extensions of score matching, Computational Statistics & Data Analysis 2007

---

### Note · Authors · 2025-08-13

Dear AC and Reviewers:

We thank the reviewers for their constructive feedback and active engagement in the discussion phase and summarise the key points:

**Summary of review comments and interactions during rebuttal**
- Reviewer VjUr found our results surprising and thought-provoking. As part of the rebuttal and discussions, we __clarified their concerns__ about comparisons against the DDPM and DDIM baselines, additional theoretical results about convergence and the computational overhead.
- Reviewer 8wzm appreciated the clarity, motivation and implementation. They raised concerns that the approach could be reduced to “diffusion models in the log domain” and about experimental validation. During the rebuttal, we __provided evidence__ that GBM is indeed not log domain diffusion, __performed__ additional experiments to compare against DDPM and DDIM, and __clarified__ the biological relevance. The reviewer acknowledged the differences between GBM and log-domain diffusion and indicated a score increase conditional on justifying our work's novelty and impact. We believe our response fully addresses this concern.
- Reviewer wtSy found the work interesting. During the rebuttal, we __reframed__ the motivation, established the link with Mirror Langevin Dynamics, __provided__ convergence analysis, __added__ new experiments with DDPM, DDIM and on CIFAR-10. The reviewer indicated that they were leaning towards increasing their score, and we believe that we’ve convincingly clarified their concerns.
- Reviewer KkH2 found the proposed framework interesting and novel. We __addressed__ the questions on convergence, biological motivation and additional experiments to compare against DDPM and DDIM.

**Summary of the contributions**
- We developed a novel generative framework that relies on multiplicative noise model.
- We established connections between Dale’s law and geometric Brownian motion, which offer various insights into developing multiplicative updates for sampling.
- We proposed a biologically-inspired sampler and provide a theoretical framework to derive this using geometric Brownian motion.
- We also developed a new formalism for multiplicative denoising score-matching which provides a firm theoretical basis for Hyvaerinen’s heuristic proposition to perform explicit score-matching for non-negative data.

We hope that the above summary of the results and interactions during the discussion will enable the AC and reviewers in their decision-making process.

---

### Decision · Program_Chairs · 2025-09-17

**Decision:**

Reject

**Comment:**

This paper proposes a variant of diffusion models driven by geometric Brownian motion. This leads to diffusions that are positive and the authors draw connections between this positivity constraint and Dale's law, which sign constrains synaptic weights in biological neural networks. The main concern that was raised by reviewers is that the paper does not represent a significant advance. Two of the reviewers found the paper conceptually novel; however, the other two reviewers thought that the paper does not represent a significant advance. Even after further engaging with the authors, the two reviewers remained unconvinced of the work's novelty. Therefore, I recommend rejection.